# Swimming behavior and hydrodynamics of the Chinese cavefish *Sinocyclocheilus rhinocerous* and a possible role of its head horn structure

**Fakai Lei**[1], **Mengzhen Xu**[1]*, **Ziqing Ji**[2], **Kenneth Alan Rose**[3], **Vadim Zakirov**[4], **Mike Bisset**[5]

**1** State Key Laboratory of Hydroscience and Engineering, Tsinghua University, Beijing, China, **2** State Key Laboratory of Hydraulic Engineering Simulation and Safety, Tianjin University, Tianjin, China, **3** Horn Point Laboratory, University of Maryland Center for Environmental Science, Cambridge, MD, United States of America, **4** Commercial Space Technologies, Ltd., Hanwell, London, United Kingdom, **5** Department of Physics, Tsinghua University, Beijing, China

* mzxu@mail.tsinghua.edu.cn

**Data Availability Statement:** All relevant data are within the paper and its Supporting Information files.

## Abstract

The blind troglobite cavefish *Sinocyclocheilus rhinocerous* lives in oligotrophic, phreatic sub-terranean waters and possesses a unique cranial morphology including a pronounced supra-occipital horn. We used a combined approach of laboratory observations and Computational Fluid Dynamics modeling to characterize the swimming behavior and other hydrodynamic aspects, *i.e.*, drag coefficients and lateral line sensing distance of *S. rhinocerous*. Motion capture and tracking based on an Artificial Neural Network, complemented by a Particle Image Velocimetry system to map out water velocity fields, were utilized to analyze the motion of a live specimen in a laboratory aquarium. Computational Fluid Dynamics simulations on flow fields and pressure fields, based on digital models of *S. rhinocerous*, were also performed. These simulations were compared to analogous simulations employing models of the sympatric, large-eyed troglophile cavefish *S. angustiporus*. Features of the cavefish swimming behavior deduced from the both live-specimen experiments and simulations included average swimming velocities and three dimensional trajectories, estimates for drag coefficients and potential lateral line sensing distances, and mapping of the flow field around the fish. As expected, typical *S. rhinocerous* swimming speeds were relatively slow. The lateral line sensing distance was approximately 0.25 body lengths, which may explain the observation that specimen introduced to a new environment tend to swim parallel and near to the walls. Three-dimensional simulations demonstrate that just upstream from the region under the supra-occipital horn the equipotential of the water pressure and velocity fields are nearly vertical. Results support the hypothesis that the conspicuous cranial horn of *S. rhino-cerous* may lead to greater stimulus of the lateral line compared to fish that do not possess such morphology.

**Funding:** The funder (the National Natural Scientific Foundation of China) only provide financial support for the research, and the authors Fakai Lei and Mengzhen Xu from the funders (The State Key Laboratory of Hydroscience and Engineering and The Tsinghua University) carried the study design, data collection and analysis, decision to publish, or preparation of the manuscript.

**Competing interests:** VZ is employed by Commercial Space Technologies Ltd. There are no patents, products in development or marketed products to declare. This does not alter our adherence to PLOS ONE policies on sharing data and materials.

## Introduction

Over 150 species of cavefish have been discovered in China, accounting for over one third of the total number of cavefish species recorded worldwide [1–3]. Many cavefish species are at-risk or threatened, and functional extinctions likely have occurred or will occur even before numerous species are officially described [4, 5]. The overwhelming majority of Chinese cavefish species are classified as belonging to the genus *Sinocyclocheilus* (family *Cyprinidae*, sub-family *Barbinae*). *Sinocyclocheilus* cavefish display a stunning spectrum of distinctive adaptations to hypogean life unequaled by any other monophyletic cavefish group [6, 7].

*Sinocyclocheilus* cavefish can be subdivided into troglobite and troglophile species [8, 9]. This division is not totally phyletic [7, 10]. Troglobite species exhibit pigment loss (which is sometimes partially-reversed by prolonged exposure to light), blindness (eyes degenerate or completely lost), reduction or total absence of scales, and often dorso-ventral stretching coupled with a pronounced bump in the region of the dorsal juncture of the head and the body [11, 12]. In several cases, including for *S. rhinocerous*, a generally anterior-pointing horn projects from this region (in which case the hump takes on the general appearance of a frill-like buttress). The humps and/or horns are components of the overall sculpted morphological form of the anterior and lateral body surfaces in some troglobite species such as *S. rhinocerous* (see Fig 1A). The anterior nasal region is nearly flat; then, moving posteriorly, the dorsal surface slopes upward. *S. rhinocerous* in particular has tiny non-functioning eyes, only a few scales along the lateral line, and lacks pigment (unless exposed to light for an extended period).

Troglophile species are also always associated with caves, though they may venture out of caves more often than troglobites. Troglophiles also exhibit marked adaptations to hypogean environments. They typically have pigment, eyes, bodies are hypertrophied in size and covered with particularly-fine scales (for a Barbinae species). Further, they are more elongated and streamlined with no dorso-ventral stretching, and they generally completely lack the cephalic humps or horns so conspicuous in troglobites. Superficially, troglophiles more closely resemble surface-dwelling (epigean) cyprinids (e.g., *S. angustiporus*–see Fig 1C).

The anterior body outline of the troglobites seemingly entails an energetic cost due to increased drag as compared to the more streamlined form typical of the troglophiles and epigean Barbinae. On its lateral body surfaces, *S. rhinocerous* possesses an extensive array of superficial neuromasts that extend from immediately behind the head until near the base of

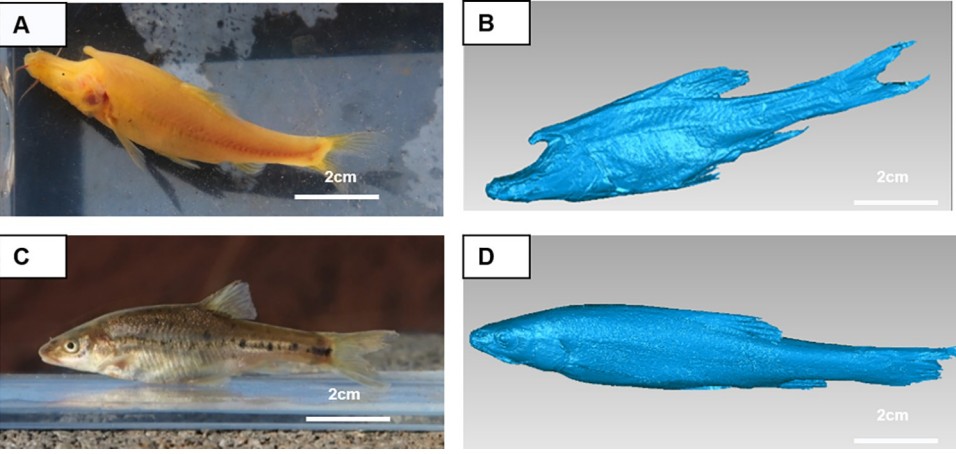

**Fig 1. *S. rhinocerous* and *S. angustiporus* individuals.** (A) A *S. rhinocerous* individual, and (B) a 3-D digital model of a *S. rhinocerous* specimen. (C) A *S. angustiporus* individual, and (D) a 3-D digital model of a *S. angustiporus* specimen.

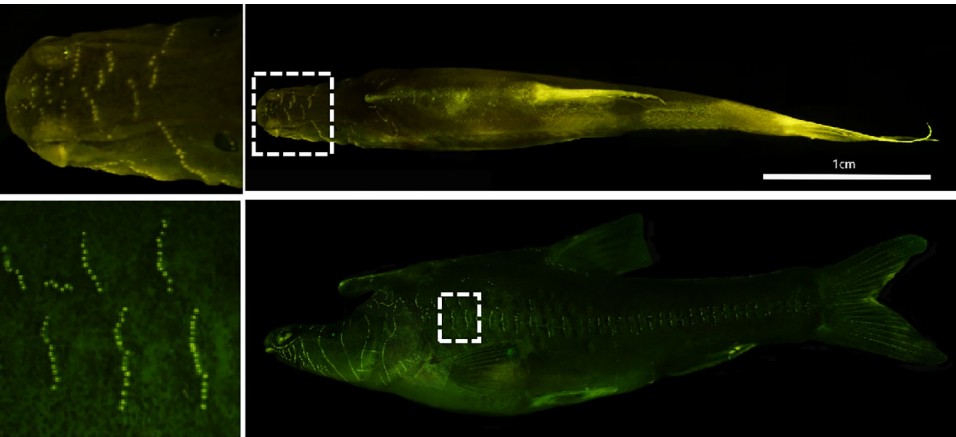

**Fig 2. Superficial neuromasts distribution of *S. rhinocerous*.** The yellow dots are the superficial neuromasts obtained by DASPEI stain. The upper two photographs show a dorsal view and the lower two show a side view of the *S. rhinocerous*.

the tail (Fig 2). These neuromasts are roughly uniformly-spaced along more-or-less regularly-spaced vertical lines extending (more dorsally than ventrally) from the mid-body lateral line, with additional smaller disconnected line segments around the dorsal fin and the head-body juncture. This extensive array of superficial neuromasts is in addition to the more-typical facial and lateral line canal neuromast systems (for morphology of lateral line system, review, see [13–15]; for mathematical analyses of lateral line sensitivity, see [16–18]).

The approach herein is to combine results from live-specimen laboratory experiments with those from Computational Fluid Dynamics (CFD) modeling to describe both cavefish swimming behavior and water flow patterns around the fish's body, with the aim of elucidating the role played by the neuromast sensory system [19]. Swimming dynamics are crucial to many facets of fish behavior including food capture/acquisition [20, 21]. Understanding swimming capabilities can also help assess the capacity of individuals to cope with variation in habitat conditions [22, 23].

The most studied cavefish, the eyeless, unpigmented Mexican cave characin, *Astyanax mexicanus* (known in older literature as *Anoptichthys jordani*) is represented by only a single species (of which a number of genetically-distinct cave-dwelling populations have been identified). It also has a surface-dwelling form (with pigment, well-developed eyes, *etc*.). Several studies have examined the differences, including in neuromast distributions, between epigean and hypogean forms [24–28]. There have been numerous observational, live-specimen studies of the sensory capabilities of the Mexican cave characin [15, 29–43]. An analysis utilizing CFD modeling was also performed [40, 41] wherein the shape of the fish was represented by a simplified aerofoil design (NACA 0013).

Thus far, there has been no dedicated study of the swimming or hydrodynamic properties of any of the troglobite *Sinocyclochelus* with their unique morphologies (indeed, apparently there has never been a CFD study utilizing a realistic model for a cavefish's body based on a scan of an actual specimen.). And, as has been noted elsewhere (*e.g.*, [1]), the function of the humped back and the supra-occipital horn, or 'head-horn' for short, has been the subject of considerable speculation. We purport to provide support for the hypothesis that these morphological features influence the hydrodynamic flow of water along the fish's body during swimming, and in so doing they act to increase the stimulus of the neuromast sensory organs to the fish's nearby environment.

A combined approach of laboratory observations and CFD modeling was applied. By combining the laboratory measurements with CFD modeling, a more complete depiction of the fluid dynamics, pressure distribution, and shear distribution along the fish body can be obtained. CFD modeling (utilizing ANSYS software) was performed on digital models of the troglobite *S. rhinocerous* and, for comparison, the troglophile *S. angustiporus*. These models were generated from scans of preserved specimens. This is in contrast to earlier studies that approximated the fish's body by a few chosen basic, axially-symmetric (along the anterior to posterior axis) geometric shapes [17, 40, 41]. Windsor *et al.* [40, 41] used a (two-) three-dimensional NACA 0013 aerofoil to obtain a (2-D) 3-D model for the surface of the fish's body. In the present study, laboratory and simulation results are combined to chart 3-D trajectories and map out velocity fields, and these are employed to elucidate the fish's swimming behavior. Mapping out the flow field around the fish enables an estimation of drag coefficients and lateral line sensing distances. The discussion and conclusion contain a justification of the assertion that the unique morphological features of troglobites such as *S. rhinocerous* influence neuromast perception, and describe some possible next steps.

## Materials and methods

### Ethics statement

All collection and laboratory measures for the fishes in this study were inspected and approved by Kunming Institute of Zoology (KIZ), Chinese Academy of Science (approval Number: SYDW-2014020). All researchers had received appropriate training and affirmed before conducting animal studies.

### Study area and cavefish specimens

Most Chinese *Sinocyclocheilus* cavefish inhabit caves in the Yunnan-Guizhou Plateau, a well-developed karst landscape. The adult specimens of *S. rhinocerous* had a maximum body length of 10 cm (general body length 4–10 cm) (Fig 1A). The sampled fish were transported and then maintained in the Endangered Fish Conservation Center (EFCC) of the Kunming Institute of Zoology (KIZ), Chinese Academy of Sciences (CAS). In EFCC we had the opportunity to observe over 10 healthy and freely-swimming specimens of *S. rhinocerous*, and were allowed to use a representative specimen in our flume experiments for over one week.

The specimens that died during the transportation and maintenance at the facility were preserved in 75% alcohol solution, and the biggest one sampled was scanned for imaging and subsequent digital modeling (Fig 1B and 1D) by the Insight3 (Open Technologies) instrument at high scanning resolution (<0.05 mm per pixel). This digital model was later used in CFD simulations to investigate the hydrodynamics of water flow around the fish.

### Experimental design

Two sets of laboratory experiments were performed to investigate the swimming behavior, flow fields, drag coefficients, and sensing distances of *S. rhinocerous* (Fig 3). Experimental conditions met the species' general habitat requirements, with water temperatures maintained at $17 \pm 0.5°C$ and exposed to ambient light (*S. rhinocerous* does not react to light). Specimen used in the experiments had an body length (BL) of 8.6 cm, body height (BH) of 2.2 cm, body width (BW) of 1.1 cm, body surface area (A) of 36.6 cm$^2$, body volume (V) of 4.4 cm$^3$, and mass (M) of 4.84 g.

The laboratory research consisted of observations of fish swimming in an aquarium. This utilized a motion capture and tracking system coupled to a Particle Image Velocimetry (PIV)

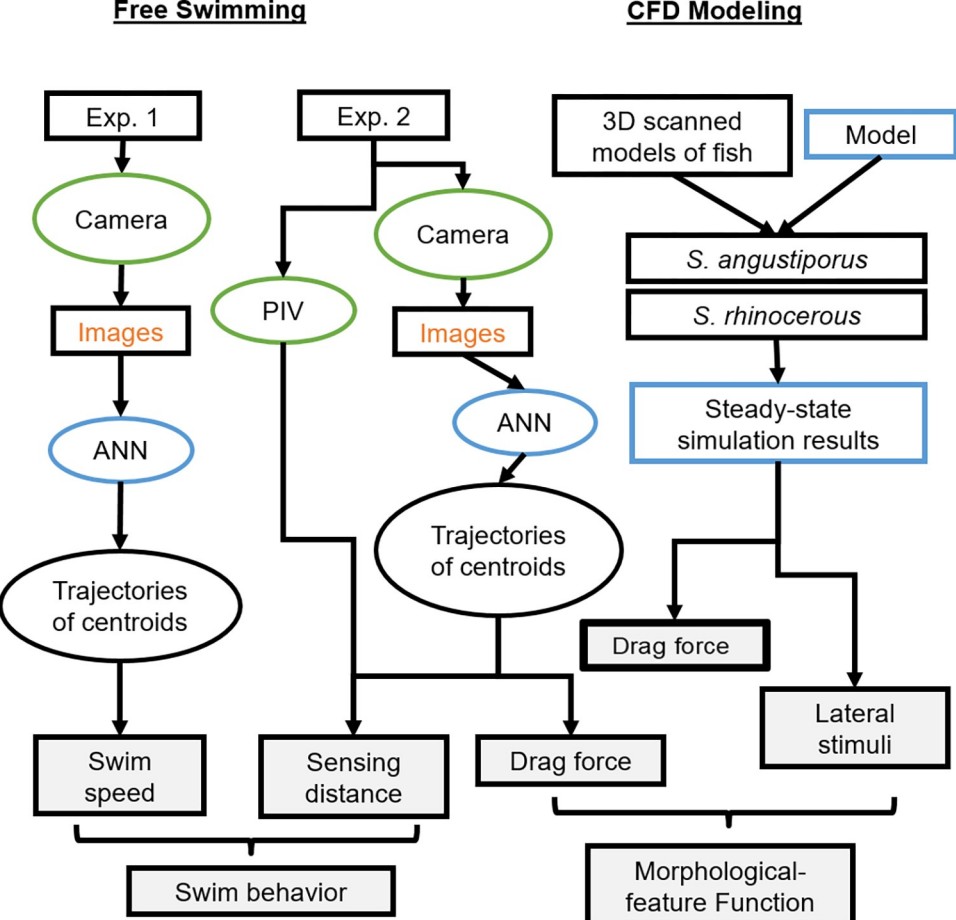

**Fig 3. Technical route.** Schematic depicting the experimental and analysis steps in Experiments 1 and 2 with free-swimming fish and the CFD modeling using models created by scanning preserved specimens. Gray boxes show the final response variables used in analyses.

system to map out water-velocity fields. A tracking system with an Artificial neural network (ANN), Convolutional Networks for Biomedical Image Segmentation (U-Net) [44, 45] was utilized to generate accurate trajectories (Patent 202010478461.5). A PIV system was also used to obtain qualitative and quantitative descriptions of the flow fields around the fish body during swimming. Such flow fields can be used to estimate the energetic costs of swimming behaviors [46], pressure distributions [40, 41], thrust impulse [47, 48], and active drag [49].

Experiment 1 was designed to observe the free swimming behavior of *S. rhinocerous* when introduced into a new environment and to estimate the average swim speed (Fig 3). Three 3-hr trials were performed on the same fish. It was found that the fish behaved differently in the first two hours and became stable in the third hour, thus we analyzed the swimming behavior during the first two hours. For each trial, we placed the fish in a 40 × 25 × 8 (cm) water-volume tank (length x width x height) equipped with a motion capture and tracking system (Fig 4, cameras 1 and 2: on; camera 3 and laser: off) and the 3-D locations of the fish were recorded. Using cameras 1 and 2 resulted in data collected at 50HZ.

Experiment 2 was designed to observe the swimming trajectories of the fish and to enable measurements of the flow field around the fish. The latter were used to empirically-estimate the drag coefficient and the potential maximum sensing distance (Fig 3). Six 3-hr replicate

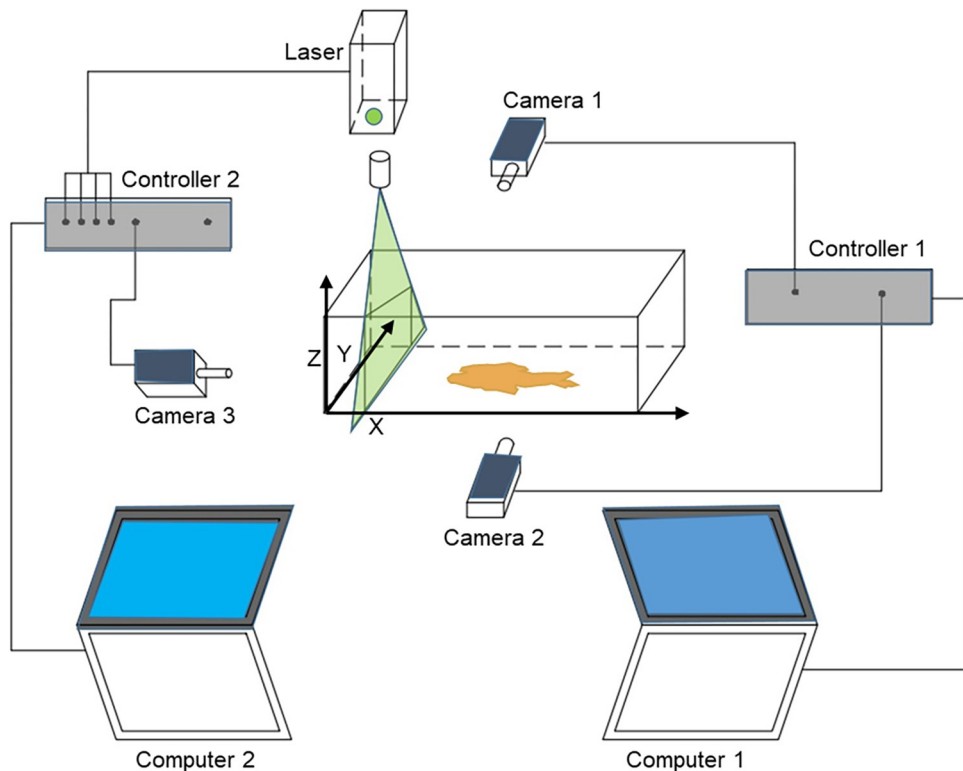

**Fig 4. Schematic drawing of the camera tracking and recording setup used in laboratory experiments.** The trajectory tracking system consisted of three cameras, Cameras 1 and 2 (Canon 80D, 50Hz) and Camera 3 (10HZ), a PIV Laser (200HZ), and two controllers and two computers (Dell XPS13).

trials were performed on the same fish; during each trial we placed the fish in a $40 \times 8 \times 8$ (cm) water-volume tank with *in situ* tiny polystyrene tracer particles (with diameter 10–100 μm) and allowed to acclimate for two hours. All measurements were then made during the third hour. As with Experiment 1, the 3-D locations were recorded. In addition, PIV was used to estimate 2-D velocity fields five times per second in the longitudinal (x-z) plane and transverse (y-z) plane. As the fish swam, PIV measurements were taken during the steady or coasting phase of swimming. A PW laser was turned on, PIV model, blue light, sheet laser, 32×32 pixel size were selected (Lifang Technique company, China) with the fish crossing the laser plane multiple times. Inclusion of camera 3 limited the resolution of the collected data to 10HZ. The program MicroVec V2.0 System (Tianjin University) was used to generate 2-D velocity fields.

**Image processing (Experiments 1 and 2).** In Experiments 1 and 2, the raw 3-D location data acquired by the fish tracking system was converted into space-time trajectories for the centroid of the fish's locations (see S1 Appendix). All top-view and side-view videos were separately converted into photo images. The trajectory of the fish's movement was then estimated using the tracking system and by recording (50 times per second) the x, y, and z coordinates of the centroid.

The instantaneous speed $u_n$ (cm/s) of the fish at time $t_n$ is determined from the ANN-generated centroid coordinates by averaging over two sequential displacements centered on the time $t_n$:

$$u_n = \left( v_{n,x}{}^2 + v_{n,y}{}^2 + v_{n,z}{}^2 \right)^{1/2}, \tag{1}$$

where

$$v_{n,x} = \frac{|x_{c,n+1} - x_{c,n}| + |x_{c,n} - x_{c,n-1}|}{2\Delta t}, \tag{2}$$

$$v_{n,y} = \frac{|y_{c,n+1} - y_{c,n}| + |y_{c,n} - y_{c,n-1}|}{2\Delta t}, \tag{3}$$

$$v_{n,z} = \frac{|z_{c,n+1} - z_{c,n}| + |z_{c,n} - z_{c,n-1}|}{2\Delta t}, \tag{4}$$

here $(x_{c,n-1}, y_{c,n-1}, z_{c,n-1})$, $(x_{c,n}, y_{c,n}, z_{c,n})$, and $(x_{c,n+1}, y_{c,n+1}, z_{c,n+1})$ are the centroid coordinates at times $t_{n-1}$, $t_n$, and $t_{n+1}$, respectively, and $\Delta t$ is time interval $(t_n - t_{n-1})$ between two adjacent frames, which is 1/50 s.

In Experiment 1, the first two hours of each trial generated over a million values for $u_n$ (50/s × 3600 s/hour × 2 hours/trial × 3 trials = $1.08 \times 10^6$). These were placed into four sequential 30-minute bins (0–30, 30–60, 60–90, and 90–120 min from onset of experiment). For each time $t_n$, the distance to the closest tank wall was also calculated.

In Experiment 2, two sets of photo images were selected from among all those extracted from the videos. Set 1 consisted of 65 series of sequential images (mean duration of a series was 2.2 s, series durations varied from 0.47 to 5.2 s) each depicting coasting swimming behavior. Set 1 was used to estimate the drag coefficient during coasting. Set 2 consisted of 21 series of sequential images (mean duration of a series was 8.6 s, series durations varied from 6 to 12 s) with each including three behavioral stages: (1) fish swimming toward the laser plane, (2) fish's head approaching the laser plane, and (3) fish swimming away from the laser plane. Each image was matched to the concurrent water velocity fields obtained from the PIV. Set 2 was used to estimate a maximum sensing distance.

**Drag coefficient (Experiment 2).** The 65 series of images depicting coasting behavior in Set 1 were used to estimate the drag coefficient of swimming. *S. rhinocerous* swimming exhibits the burst and coast swimming behavioral style common to many fish species [20, 49, 50]. In the burst phase, the tail moves with one or two strokes to accelerate; this is followed by the coasting phase in which the fish rests (glides) with its body straight. A Reynolds number (Re) of 1280 was estimated for the coasting phase, indicating that the drag force is proportional to the square of the speed (see S2 Appendix). The drag coefficient was calculated using (S2 Appendix):

$$\frac{1}{u(t)} = \frac{C_{d,coast}\rho A}{2(m + k\rho V)} t + c, \tag{5}$$

where $\rho$ is the water density (kg/m$^3$), A is the fish's surface area (m$^2$), V is body volume (m$^3$), $k$ is a mass coefficient set to 0.045 [51, 52], and $c$ is a constant from fitting. For each of the 65 image series, linear regression was applied to the observed swimming speeds over time, and a value for the drag coefficient ($C_{d,coast}$) was derived from the slope coefficient. This yielded 65 estimates for $C_{d,coast}$.

**Potential maximum sensing distance (Experiment 2).** For each of the 21 series of images in Set 2, values for the flow's kinetic energy at plane of the laser ($E_L$) over sequential 0.2 second time intervals were computed from the PIV-generated water velocity field via:

$$E_L = \int_0^s \frac{1}{2}\rho h v^2 \, ds, \tag{6}$$

where h = 2 mm is a constant attributed to the 'thickness' of the laser plane, S is the area of the laser plane, and $v$ is the flow speed, *i.e.*, the particle velocity from the PIV field in the laser plane. Eq 6 integrates water velocities (from the particles) over the area of the laser plane at a given time. Each $E_L$ value was then matched with the centroid location (and the derived speed) at the midpoint time of the series of images. The distance of the fish's head from the laser plane at which the value $E_L$ increases abruptly is defined as the potential maximum sense distance ($D_L$). Values for $D_L$ and the speed ($u_L$) were obtained for each series.

## CFD simulation

Preserved specimens of *S. rhinocerous* and *S. angustiporus* were scanned to generate 3-D models (Fig 1B and 1D, respectively) which were used (1) to simulate the flow field around the fish body during coasting swimming in open water and (2) to calculate both the pressure-related stimulus to the lateral system and the drag force like previous study [53]. For model simulations, this was a simulated flow that was continuously on and not a real flow in a flow channel, and the fish's body was aligned such that the head faced the incoming flow. This minimized effects due to the tail and was taken to represent cruising behavior. The CFD control volume was set to be ten times the fish's body size (10BL×10BW×10BH) to ensure that the flow field was fully developed. The fish was placed at the center of the calculation zone. Mesh size was varying from about 0.0025 m near the fish's body to 0.01 m elsewhere within the domain.

The CFD simulations were run for steady-state flow with Re ranging from about 900 to 10,000. This corresponds to swimming velocities ranging from 1 to 10 cm/s, matching values found in Experiment 1. Note that in the simulations, fish were taken as stationary while the water flowed; by contrast, in the live-specimen experiment, the water was taken to be still while the fish was moving. All physical parameters of the CFD modeling were set to either standard values, values representative of the environmental conditions in the experiments, or to values typically observed in fish swimming in Experiments 1 and 2. For each *Re* value and associated inflow velocity and swim speed, a simulation was done and steady-state velocities and pressure values were generated throughout the grid, including along the fish's body. All simulations were done using the software ANSYS FLUENT. Details of the CFD modeling are described in S3 Appendix. Model results for pressure fields were used to estimate the drag forces and the pattern of pressure-related stimuli along the body relative to the system of neuromasts.

**Drag force estimation.**   The drag force *F* for the model fish was obtained by integrating the pressure field *P*, as determined by the CFD simulation, over the component of fish's surface in the direction of the water flow. The estimated drag coefficient $C_d$ was then calculated from the drag force:

$$C_d = \frac{F}{(\rho/2)AU^2}. \tag{7}$$

The drag coefficient values obtained in the CFD simulations for *S. rhinocerous* and *S. angustiporus* were compared to those estimated for *S. rhinocerous* in Experiment 2 and to values reported in the literature.

**Stimulus to lateral line system.**   The pressure field *P* at a given location on the fish's surface, obtained via CFD simulation, was normalized by calculating a coefficient of pressure $C_p$:

$$C_p = \frac{P}{(\rho/2)U^2}, \tag{8}$$

where *U* is the inflow velocity in the simulation. This was taken as a measure of the stimulus to

a hypothetical canal pore at that site, and the difference $\Delta C_p$ between adjacent pore sites was then used to quantify the stimulus to a canal neuromast of the lateral line [54–56]. Canal pores were assumed to be spaced at 2% body length (BL) intervals as illustrated on the scanned model cavefish body. This pore interval is consistent with an earlier study of Mexican cave characins [57], and the general positioning of the pores on the *S. rhinocerous* body is in agreement, albeit idealized and approximate, with observations of live and stained specimens. More quantitative evaluation of canal pore spacing to compare to other species using our specimen was not supportable. Neuromasts appear bright yellow after stained by immersion in DASPEI, that is, 2-(4-(dimethylamino)styryl)-N-ethylpyridinium iodide (DASPEI, Invitrogen) [see 58 for the staining method]. As a consequence, $\Delta C_p$ at each assumed pore location is plotted against the distance (along the fish's surface) of that pore from the anterior end. Curves for $\Delta C_p$ *vs.* distance from anterior end generated for *S. angustiporus* and *S. rhinocerous* were compared with particular attention given to the head region. Earlier results from using simple aerofoil shapes as model fish bodies are also included for comparison [17, 40]. The symbols for the different items are listed in Table 1.

## Results

### Swimming behavior (Experiments 1 and 2)

During Experiment 1, *S. rhinocerous* prefer to swim near a wall during the first hour and then swim seemingly randomly throughout the tank during the second hour (Fig 5A). The trajectories were closer than 0.25BL to a wall for about 70% of time for the first hour (Fig 5B), while

**Table 1. List of symbols.**

| Symbol | Description | Units | Value(s) |
|---|---|---|---|
| BL | Fish body length | cm | 8.6 |
| BH | Fish body height | cm | 2.2 |
| BW | Fish body width | cm | 1.1 |
| A | Fish body area | cm$^2$ | 36.6 |
| V | Fish body volume | cm$^3$ | 4.4 |
| M | Fish body mass | g | 4.84 |
| $v_{n,x}$, $v_{n,y}$, $v_{n,z}$ | Fish velocities in 3-D | cm/s | |
| $u_n$ | Fish speed from centroids | cm/s | |
| Re | Reynolds number | | |
| $\rho$ | Density of water | kg/m$^3$ | 991 |
| k | Additional mass coefficient | | 0.45 |
| c | Intercept from regression | | |
| $C_{d,coast}$ | Drag coefficient | | |
| h | Thickness of laser plane | mm | 2 |
| S | Area of laser plane | m$^2$ | |
| v | Particle velocity | m/s | |
| $E_L$ | Kinetic energy of flow | | |
| $D_L$ | Distance from fish head to laser plane | BL | |
| $u_L$ | Fish speed at same time as $D_L$ | cm/s | |
| F | Drag force of model fish | N | |
| P | Pressure field | Pa | |
| U | Inlet water velocity | m/s | |
| $C_p$ | Coefficient of pressure | | |
| $\Delta Cp$ | Difference in coefficient of pressure | | |

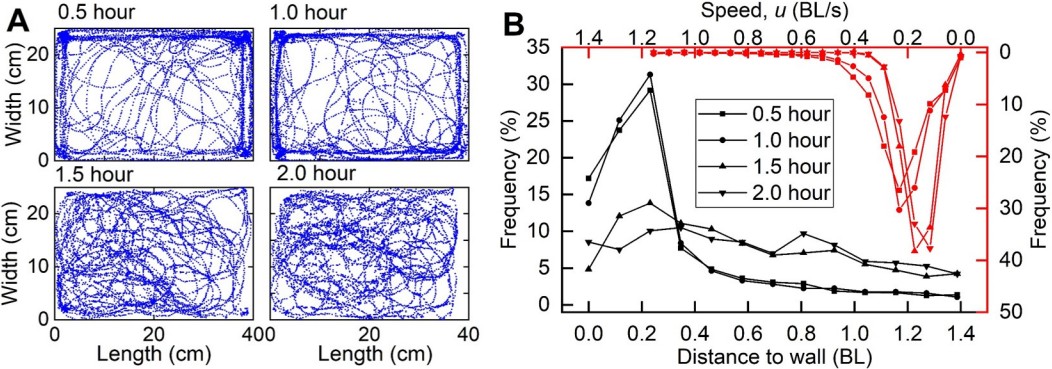

**Fig 5. Fish behaviors.** (A) Trajectory of *S. rhinocerous* projected into a horizontal plane during one trial for, 0–0.5 hour, 0.5–1 hour, 1–1.5 hours, 1.5–2 hours from onset of experiment. (B) Frequency distributions of the distance to a wall (black line) and swimming speed (red line) for the same four time bins.

the time spent near a wall decreased to about 35% during the second hour. Typical swimming speeds were from 0–0.4 BL/s (0–3.5 cm/s), and swimming speed gradually slowed with time (from an average of 2.4 cm/s during the first hour to 1.4 cm/s during the second hour).

The 65 series of images selected in Experiment 2 confirmed that *S. rhinocerous* exhibited a burst-and-coast swimming style (an example of one of these sequences is shown in Fig 6). A burst-and-coast sequence generally lasted for 3.0±0.68 s (*mean*±SD) and was comprised of a short burst (0.81sec±0.29 s) followed by a longer coast (2.15±0.63 s). The swimming speed during a burst-and-coast event varied from 0 to 10 cm/s, with an average swimming speed of 1.57 cm/s, which corresponds to Re = 1280. The maximum speed (10 cm/s) was reached in only ~1% of the sequences. In the burst phase, *S. rhinocerous* utilized fishtailing to accelerate [59], the fish moves the tail slowly to one side, followed by two fast flicks of the tail, whereupon the fish restores its straight posture and goes into the coasting phase, glided forward and gradually slowed down.

## Drag coefficient (Experiment 2 and CFD simulation)

Fig 7A shows a drag coefficient estimation example, which plotted $\frac{1}{u(t)}$ versus time for one series of sequential images from Experiment 2. The estimated drag coefficient from the example in

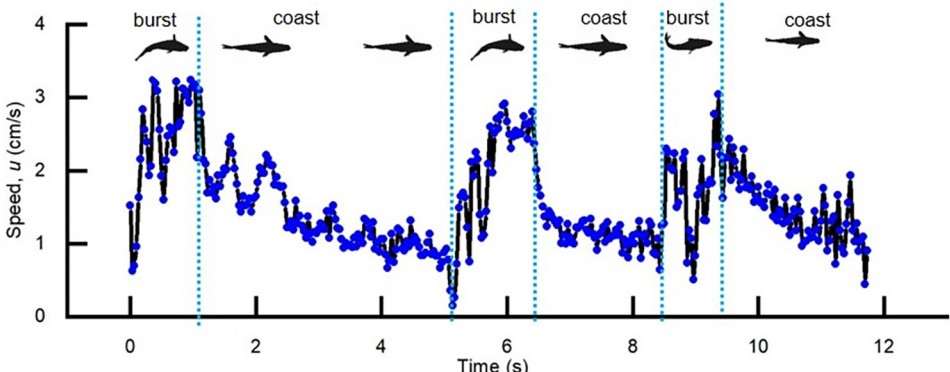

**Fig 6. Swimming speed.** Swimming speed ($u$) versus time taken from a series of images during Experiment 2. This illustrates a burst-and-coast swimming style.

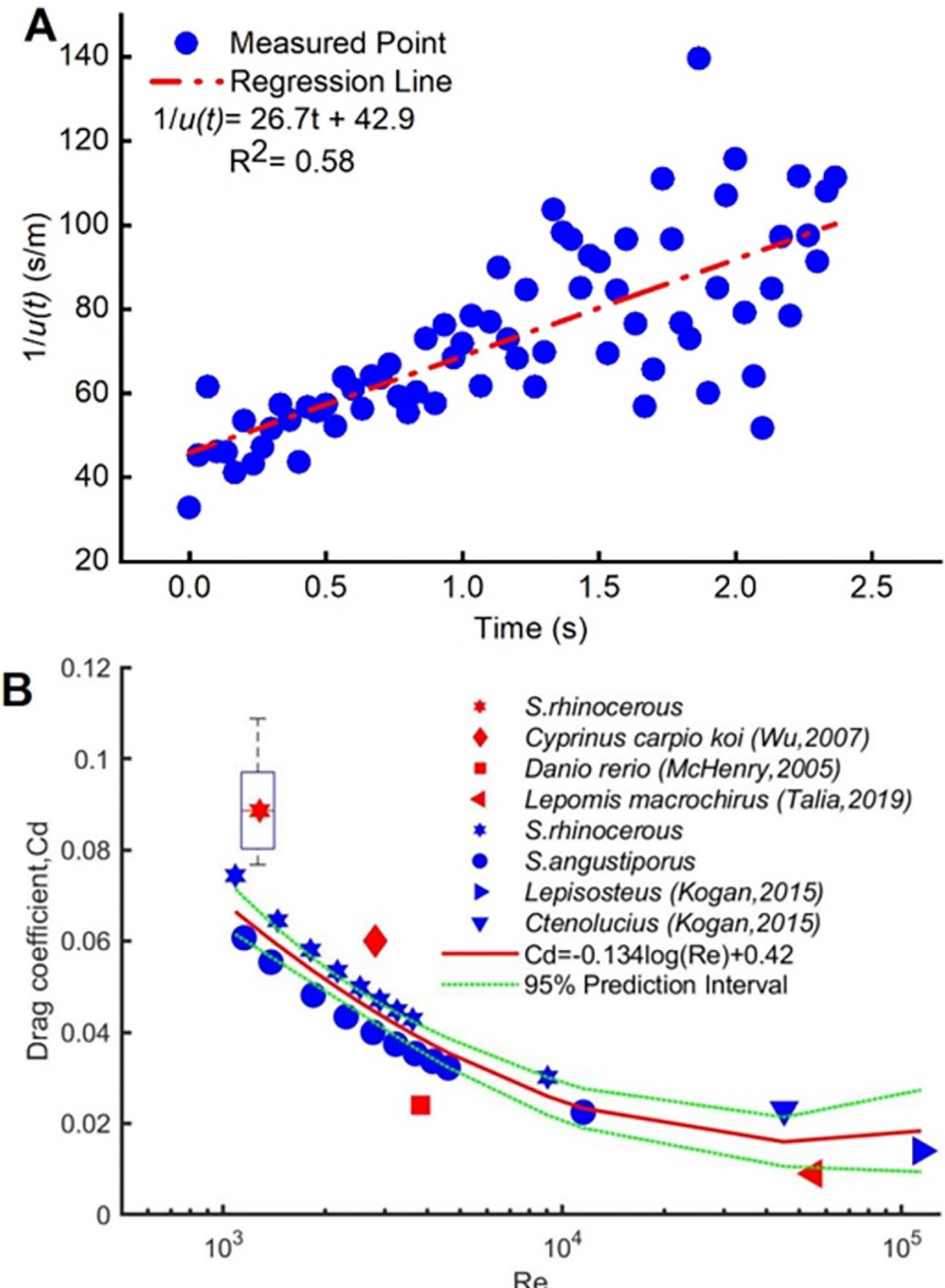

**Fig 7. Drag coefficient estimation.** (A) Reciprocal speed during the coasting phase of swimming; (B) Fish drag coefficient estimates obtained by measurements of swimming speeds (red) and by CFD modeling (blue) for analyses in this paper and reported by others. The red star indicates the median value derived for *S. rhinocerous* from Experiment 2, and the box delineates the 25th and 75th percentiles. The blue stars and circles are from the CFD simulations for *S. rhinocerous* and *S. angustiporus*, respectively. The red curve is a regressive fit to all the CFD results from this study.

Fig 7A based on Eq 5 is 0.074. Using the 65 sequences from Experiment 2, Eq 5 yielded an estimate for the drag coefficient during the coasting phase $C_{d,coast}$ of $0.09\pm0.01$ (*mean*±SD) (Fig 7B, red star). For the same Re value (assumed for the experiment), the $C_{d,coast}$ estimate obtained via Eq 7 from the CFD simulations for *S. rhinocerous* was generally consistent with that obtained from Experiment 2. The simulation estimate for $C_{d,coast}$ decreased with

increasing Re (see Fig 7B). The CFD-estimated values for $C_{d,coast}$ for *S. rhinocerous* were similar to, but consistently slightly higher than those for *S. angustiporus*. This suggests that *S. rhinocerous* requires more energy to overcome drag forces when coasting than *S. angustiporus*. Reported values for $C_{d,coast}$ based on CFD studies of other species [60] were generally consistent with the results of Experiment 2 and the CFD modeling of the *Sinocyclocheilus* species in this study. The simulation results from this study (not including the point based on Experiment 2) are fitted by the linear relation $C_{d,coast} = -0.134\log(Re)+0.42$.

## Potential maximum lateral line sensing distance (Experiment 2)

Kinetic energy ($E_L$) values obtained for each of the 21 sequences from Experiment 2 indicated that motion could in general be parsed into three stages: approaching, entering, and exiting the laser plane (Fig 8A provides an example). Typically, $E_L$ was initially low during the approaching stage (Fig 8A, stage 1), and the water velocity field was relatively smooth (Fig 8C, stage 1-M1). As the fish approached the laser plane, the $E_L$ value increased sharply, reaching a maximum value when the fish head touched (entered) the laser plane (Fig 8A, stage 2). This was accompanied by a disruption in the water velocities (Fig 8C, stage 2-M2). Finally, as the fish exited the laser plane, $E_L$ decreased and the water in the laser plane gradually resumed its initial undisturbed state as the *S. rhinocerous* individual swam far enough away from the laser plane such that the water velocity eventually decreased to values similar to stage 1 (approximately still water) pattern (Fig 8A, stage 3; Fig 8C, stage 3-M3). The ($u_L$, $D_L$) pairs collected from the 21 sequences consistently placed the sensing distance within 0.3BL, with the few high-$u_L$ points showing a greater spread (Fig 8B).

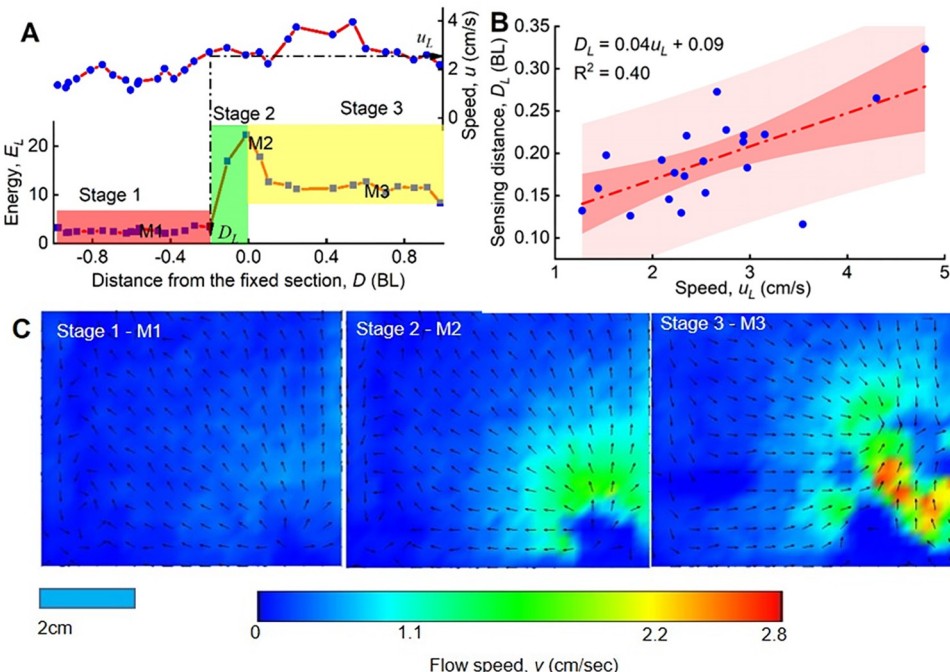

**Fig 8. Sensing distance estimation.** (A) Kinetic energy of flow ($E_L$) at the laser plane and fish swimming speed ($u_L$) in the three stages (approaching, entering, exiting laser plane) for a series of Experiment 2. The sensing distance $D$ is the distance between the fish and the laser plane; (B) Maximum sensing distance ($D_L$) versus fish swimming speed ($u_L$) showing concurrent ($u_L$, $D_L$) pairs; (C) showing representative water flow fields for a moment (M1, M2 and M3) within each of the stages.

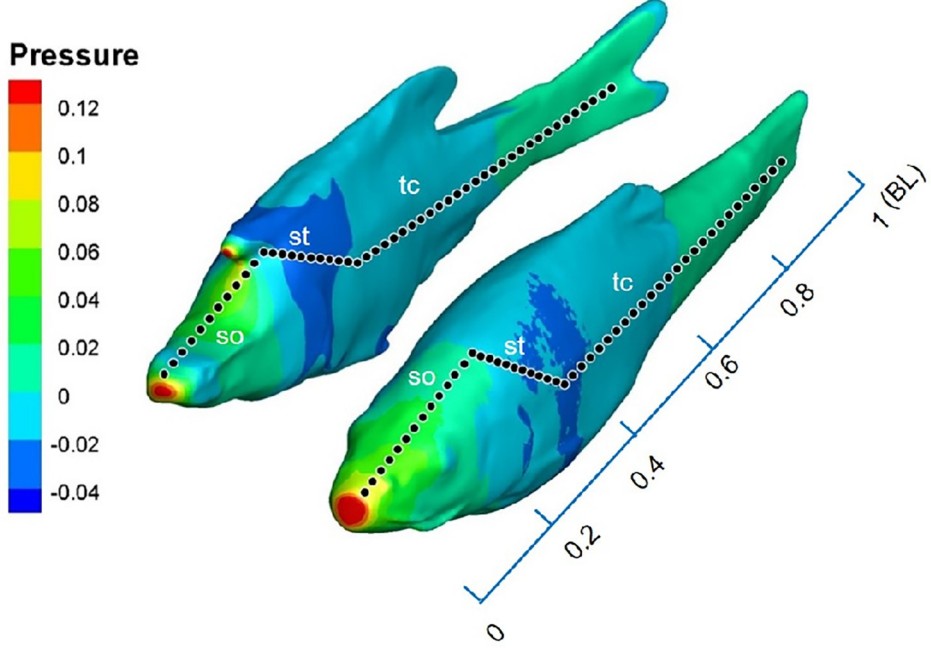

**Fig 9. Canal neuromasts distribution and pressure simulation.** The putative canal pores distribution along the fish body and CFD results showing pressure distributions of *S. rhinocerous* and *S. angustiporus* from our simulations (Re = 1280), canal pores are approximately shown as black dots with white outlines, supraorbital canal (so), supratemporal canal (sc), trunk canal (tc).

### Lateral line stimulus (CFD)

The pressure fields simulated by the CFD model exhibited a characteristic pattern for coasting fish in open water (Fig 9). Both the *S. rhinocerous* and *S. angustiporus* simulations showed small high-pressure regions at the nose as well as large low-pressure regions along the lateral body sides. One major difference was that the *S. rhinocerous* simulations showed a negative-pressure region on the head and the body close to the head (around 0–0.3BL from the anterior point), while the simulations for *S. angustiporus* did not show a pressure drop in this region.

Our CFD results on the pressure distribution along the body differed from earlier results based on simple geometric models of the fish body (see Fig 10A). Fig 10A shows the normalized pressure along the putative canal neuromasts in Fig 9. The simulated pressure distribution of *S. angustiporus* had a similar shape to that found by others using simplified axially (anterior to posterior)-symmetric models [17, 40, 41], although our simulation results showed a larger positive pressure region around the head and a lower in magnitude negative pressure region down the lateral sides (Fig 10A). A similar narrative applies for the $\Delta C_p$ results, save that in the case of *S. rhinocerous* there is a distinct jump in $\Delta C_p$ near the back of the head (Fig 10B).

### Discussion

*S. rhinocerous*, like many cavefish species, is extremely vulnerable to perturbations in its habitat [61–64]. Understanding the swimming and sensory capabilities of troglobite cavefish may help discern micro-habitat preferences and dispersal patterns of the cavefish in their underground environments [61]. In the present study, both live-specimen experiments and CFD model simulations were combined to characterize the swimming behavior and hydrodynamics of *S. rhinocerous*. For comparison, the more epigean-appearing, sympatric *S. angustiporus* was also

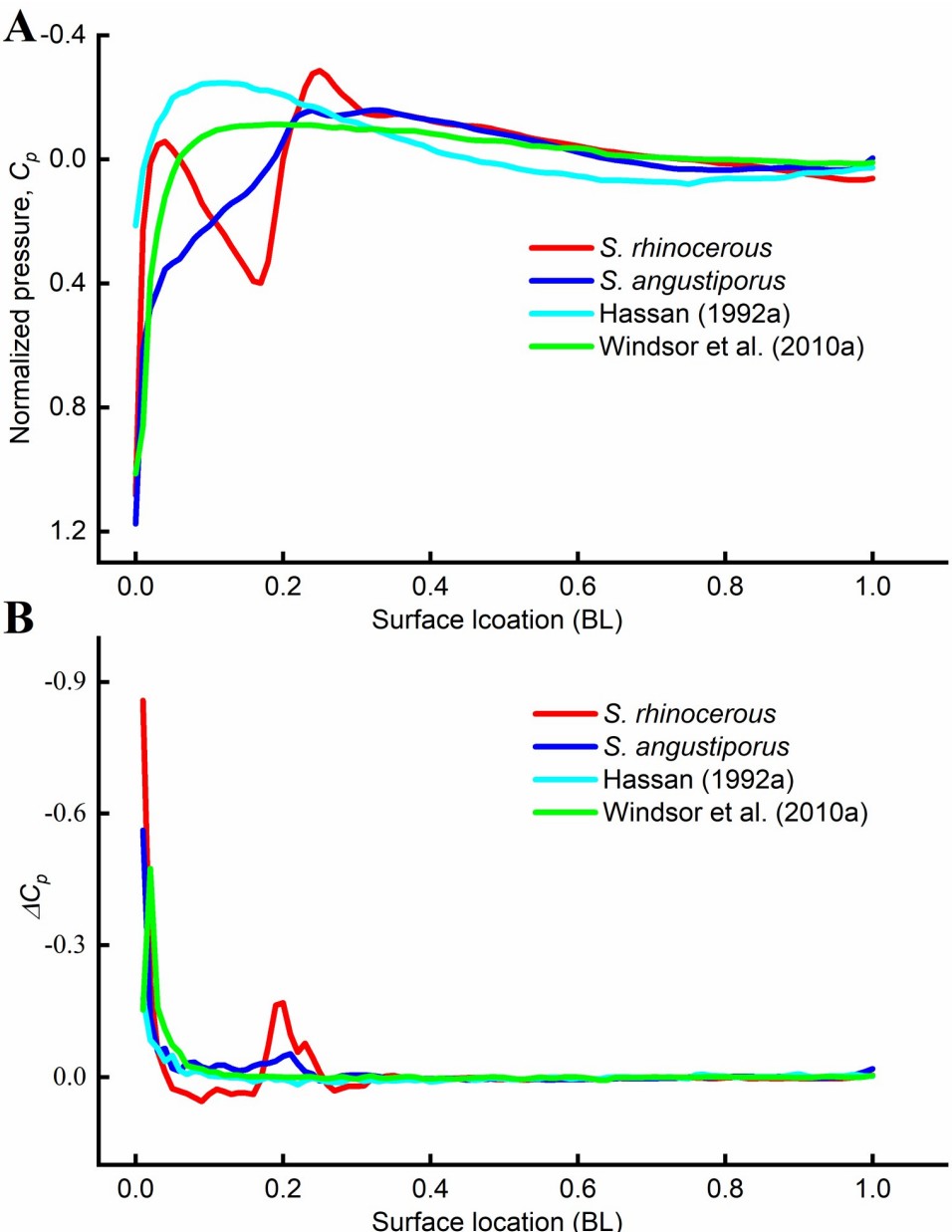

**Fig 10. Lateral line stimulus estimation.** CFD results showing (A) the normalized pressure, or $C_p$, and (B) $\Delta C_p$ values along the assumed line of canal neuromasts for the simulations. Results extracted from earlier studies are also included. The Y-axis is plotted with its scale reversed (decreasing values).

studied in the CFD simulations. Motion trajectories were catalogued, and from these important quantitative parameters including swimming speeds, drag coefficients, and others associated with sensory capabilities were derived.

## Swimming behavior

When first introduced to a new environment, *S. rhinocerous* displays near-wall-following (<0.25BL, Fig 4A) behavior, suggestive of exploratory behavior. This is followed by seemingly

random wandering throughout the tank. The sensing distance of 0.25 BL observed in the experiment (Fig 5B) is comparable to that reported for blind cave characins [40, 41], the blind hypogean morph of *Astyanax mexicanus*. Studies have examined the wall-following behavior of the blind cave characin and contrasted it with that of the sighted, epigean morph of *A. mexicanus* [65, 66]. The sighted morph only exhibited prolonged wall-following behavior in a dark environment. Compared to the blind morph, the sighted morph swam along the wall for relatively shorter distances with the head generally inclined toward the wall. When placed in an aquarium with a concave wall, the sighted morph also more frequently switched directions along the wall. Experiment 1 observations of *S. rhinocerous* were generally consistent with those noted here for the blind morph of *A. mexicanus*.

The average swimming speed of *S. rhinocerous* determined in Experiment 1 was 1.57 cm/s (Fig 5B), confirming the expectation that *S. rhinocerous* is generally a slow swimmer. Estimates of average swim speeds (obtained by a different methodology) for the decidedly-benthic amblyopsid cavefish of North America were slower [67]. In the amblopsid measurements, 4 specimens of a given species tested in an artificial stream with rock shelters were videotaped for 15-minute intervals over a period of 30 days, and the average swimming speed was estimated as 1.42 cm/s for *A. spelea*, 0.72 cm/s for *T. subterraneus*, 0.45 for *A. rosae*, and 0.16 cm/s for *F. agassizii* [67]. By comparison, *Sinocyclocheilus* troglophiles are more streamlined and are known to swim faster. For instance, during the preliminary portion of this study, the troglophile *S. qujingensis*, found around the exit of a subterranean stream, was observed in the field swimming faster than 30 cm/s. We also occasionally observed flow rates in excess of 50 cm/s near cave exits. While in general, reported fish swimming speeds vary greatly, 1.57 cm/s reported herein for *S. rhinocerous* is near the lower end of reported values [68]. The slow swimming behavior of *S. rhinocerous* may be considered an adaptation to the oligotrophic subterranean environment where both the risk from predation and the nutrient content are low, and so there is a selective advantage for conserving energy in motion. The macro-invertebrates and organic matter on which *S. rhinocerous* feeds also do not require swiftness. In addition, slow swimming behavior may affect the sensory capabilities of the lateral line system and can help to conserve energy.

Observations also demonstrated that *S. rhinocerous* swam faster when first introduced into the new environment. This could reflect an acclimation period or a reaction to stress; however, it could instead indicate an exploratory phase during which a new environment or sudden habitat change is being assessed. In the experiments, *S. rhinocerous* tended to maintain a distance from the wall in rough agreement with the estimated maximum sensing distance of the lateral line system (compare Figs 5B and 8B); we therefore hypothesize that the lateral line system plays a dominant role in how the cavefish explores a novel or altered environment.

## Drag coefficients and lateral line stimulus

The larger drag coefficient for the motion of *S. rhinocerous* gleaned from the CFD modeling as compared to that found for *S. angustiporus* (Fig 7B) suggests that the body shape of *S. rhinocerous* has lower hydrodynamic efficiency and a higher energetic cost for movement. The *S. rhinocerous* morphology creates a near-vertical upstream surface in the head region under the head-horn (Fig 9). This results in increased pressure and a higher drag coefficient.

The simulated pressure value in the location of the canal pores in Fig 9 for *S. rhinocerous* and *S. angustiporus* was extracted. The highest pressure occurs at the stagnation point at the tip of the snout (Fig 10A). Stimuli at canal pore sites in the head region are higher than at pore sites posterior to the head. Results from the *S. rhinocerous* and *S. angustiporus* simulations clearly demonstrate a noticeable impact on the distribution of the pressure stimuli at different

locations on the body surface from the two species' markedly different cephalic morphologies: the buttressed head-horn structure in its entirety enhances perception of pressure changes by the canal neuromasts (Fig 10B). We propose that the enhancement of this passive sensory system in *S. rhinocerous* more than compensates for the extra drag costs associated with locomotion (and accordingly slower typical swimming speeds), yielding an over-all increase in fitness of the troglobite in its dark, nearly predator-free habitat.

## Horn function and other horn-bearing fish species

Especially given the oligotrophic, typically pitch-dark conditions observed during surveys done as part of this study, it seems apparent that the head-horn and dorsal humps seen in *S. rhinocerous* and other *Sinocyclocheilus* troglobites must be a troglomorphic trait; however, how such morphological features are beneficial remains unclear [1, 69]. Analyses of *S. rhinocerous* and closely-related species have led to several hypotheses as to the function of the head-horn. The 3-D morphology of the head-horn in *S. hyalinus* shows a horn cavity that may offer a channel from the fenestrae in the frontal wall to the cranial cavity associated within the horn, thereby enhancing acoustic perception [69]. The role of the head horn of *S. rhinocerous* has also been suggested as protection for the head [70], as scratches on the head horn in some species in field were observed. Similar observations have also been suggested to indicate that the head-horn may be used to attach to underwater objects as a means of saving energy (Zhao Y., personal communication). It has even been suggested that the horn plays a role in fat storage [3]. Our analysis of swimming behavior and of the hydrodynamics associated with the troglobite morphology support the hypothesis that the head-horn, and more generally the unique sculpted form of the anterio-dorsal surface of *Sinocyclocheilus* troglobites (including humped dorsal surfaces and frill-like contours at the head-body juncture), benefit the sensory capabilities of the extensive array of neuromasts arrayed along the troglobite bodies.

There are a few non-hypogean fish species that also have a head-horn. For instance, Indo-Pacific surgeonfishes of the genus *Naso* (Family Acanthuridae); however, it is unlikely that the same function of *Naso* head-horn is necessary. Because *Naso* differs from *S. rhinocrous* in eyesight and superficial neuromast distribution. *Naso* keep eyesight and they don't show extensive array of superficial neuromasts. Another example is the nurseryfishes of Australia and Asia (Family Kurtidae; *Kurtus gulliveri* and *Kurtus indicus*, respectively), which live in mangrove swamps instead of caves and are sexually dimorphic with only the males possessing a horn (which is short and more like a hook). In the Australian species, it has been demonstrated that the principle reproductive purpose of the hook is as a carrier of an egg sac [71–73]. However, there is no sexual dimorphism in the cranial morphology of *S. rhinocerous* or other *Sinocyclocheilus* species.

## Combining experiments and CFD modeling

Previous CFD simulation studies have investigated how swimming behavior can affect the pressure and shear distributions along the fish body. The shear distribution on the fish's body during cyclic swimming showed that thrust is mainly produced in the posterior half of the body [74]. Simulations also have demonstrated that the peak thrust in the cycle is associated with a leading-edge vortex [75]. Simulation of flow fields from yolk-sac larvae to juveniles shows that larvae need to continuously adjust their sensory, neural, and muscular systems to adapt to changing flow regimes [76]. Furthermore, simulations coupled to the motion of fish explain the traveling wave speeds of the muscle activations [77]. Swimming kinematics are also strongly influenced by the morphology of the particular species [78]. Simulation of hydrodynamics in the nasal region of the sturgeon *Huso dauricus* suggests that swimming alone is

sufficient to drive olfactory flow, and that vortices within the olfactory flow may help transport the chemical stimuli to the sensory surfaces [79]. This may be quite significant in the case of *S. rhinocerous* since the olfactory cavity is clearly hypertrophied in this species. Additionally, the tactile barbels in *Sinocyclucheilus* are extremely long compared to other barbs (sub-family *Barbinae*, family *Cyprinidae*) and no doubt also play a key role in prey detection. It has been suggested that, compared to an enhanced system of neuromasts, hypertrophy of barbels was a more primitive developmental adjustment to the loss of vision in the cave-dwelling *Sinocyclocheilus* species [3].

CFD simulation of flow and pressure fields have previously set the stimulus to the lateral line of a gliding specimen in still water to be a dipole [80], a fixed wall [40, 41] or an approaching predator [81]. Such analyses confirmed that the flow fields in the boundary layer around the fish's body are important to lateral line system [15, 80, 82, 83]. Our study found that the canal neuromast system (lateral line system) in *S. rhinocerous* was only sensitive to flow fields over a very limited distance (~0.25 BL) from the fish's body. This is almost the same as earlier findings (~0.2 BL) for the Mexican blind cave characin [40, 41]. However, the current study's use of realistic models based on scans of preserved specimens in addition found distinctions between the troglobite *S. rhinocerous* and the troglophile *S. angustiporus* in $\Delta C_p$ values in the cephalic region (0–0.3 *BL* from the anterior end).

## Future plans

Further experimental analyses could involve a larger number of test specimens to enable a closer examination of variability among conspecific specimens. Our results are dependent on one individual fish that can limit the robustness of our results. While we are confident the behavior and morphology are representative, individuals will vary in other aspects (e.g., swimming speed) that could affect the specifics of our results. Variability within a species can also be important in assessing how cavefish populations will respond to changes in habitat [61, 84]; however, the same references cited here also caution against over-collection from small populations. Additional work on multiple individuals to assess inter-individual variability is warranted.

Additional studies are also needed to examine the acceptable range for numerous habitat parameters, including water quality, so that conservation guidelines may be established. In addition to possibly being toxic or fatal to the cavefish, changes to their environment may affect their behavior and thus negatively-impact the long-term survivability of the species. CFD modeling such as that introduced in this paper can also be used with to determine pressure and shear conditions along the fish's body, and the associated energetics, for different swimming modes and under changing environmental conditions without having to expose valuable specimens to an array of sometimes potentially harmful conditions. Such information, when added to the basic results reported here, will improve assessments of how changes in habitat may affect the *S. rhinocerous* and other troglobite cavefishes. Thus the impacts from novel stressors can be anticipated, and conservation activities directed accordingly.

One of the key advances in the simulations performed here over those of previous analyses was the introduction of realistic models for the fish's body based on scans of preserved specimens. The modeling of the canal pore locations on such models was quite reasonable, but not reaching a level that is certainly possible. That is, to use the preserved specimen to map the reasonably-exact locations of the pores on the body surface (by use of a stereoscopic microscope and mapping software such as ImageJ) onto the model. Mapping the positions of the superficial neuromasts on the body, either by staining to indicate neuromast locations [38, 85] or by microscope, will be more time-consuming, but is quite doable. Given this robust model of the

body morphology and its neuromast locations, an analysis of the potential of such a system to detect and discern random shapes or disturbances under assorted flow conditions using an analysis akin to that in Sichert *et al.*, [86] may be feasible, though reaching such a level of sophistication may require a series of intermediate steps that are coordinated with targeted experiments.

## Supporting information

**S1 Appendix. Training of ANN to estimate centroid locations.**
(DOCX)

**S2 Appendix. Derivation of drag coefficient equation.**
(DOCX)

**S3 Appendix. Details of the CFD modeling of *S. rhinocerous* and *S. angustiporus* body shapes.**
(DOCX)

**S1 File. Dataset.**
(XLSX)

**S2 File. Code share with PlosOne.**
(ZIP)

**S3 File. ANN tutorial.**
(DOCX)

## Acknowledgments

We thank Dr. Junxing Yang, Dr. Xiaoyong Chen, Dr. Li Ma and Dr. Xiaofu Pan from Kunming Institute of Zoology, Chinese Academy of Sciences for their support in experimental materials and laboratory facilities, and Hanmo Chen and Tiankai Yang from Kunming Engineering Corporation Ltd., China for their assistance during the experiments.

## Author Contributions

**Conceptualization:** Mengzhen Xu, Mike Bisset.

**Data curation:** Fakai Lei, Mengzhen Xu, Ziqing Ji.

**Formal analysis:** Fakai Lei, Mengzhen Xu, Ziqing Ji, Kenneth Alan Rose, Vadim Zakirov.

**Funding acquisition:** Mengzhen Xu.

**Investigation:** Fakai Lei, Mengzhen Xu, Ziqing Ji, Mike Bisset.

**Methodology:** Fakai Lei, Mengzhen Xu, Ziqing Ji, Kenneth Alan Rose, Vadim Zakirov, Mike Bisset.

**Project administration:** Mengzhen Xu.

**Resources:** Mengzhen Xu.

**Software:** Fakai Lei, Ziqing Ji, Vadim Zakirov.

**Supervision:** Mengzhen Xu, Kenneth Alan Rose, Vadim Zakirov, Mike Bisset.

**Validation:** Fakai Lei, Mengzhen Xu, Ziqing Ji.

**Visualization:** Fakai Lei, Mengzhen Xu, Kenneth Alan Rose.

**Writing – original draft:** Fakai Lei, Mengzhen Xu, Kenneth Alan Rose, Mike Bisset.

**Writing – review & editing:** Fakai Lei, Mengzhen Xu, Kenneth Alan Rose, Vadim Zakirov, Mike Bisset.

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
