## [Decision Letter · Decision Letter 0]

29 Jun 2021

PONE-D-21-12656

Cavefish Hydrodynamics and Behavior: A Study of the Chinese cavefish Sinocyclocheilus rhinocerous

PLOS ONE

Dear Dr. Xu,

First, I would like to sincerely apologize for the delay in the review process.

Thank you for submitting your manuscript to PLOS ONE. We feel that it has merit but does not fully meet PLOS ONE’s publication criteria as it currently stands. We invite you to submit a revised version of the manuscript that addresses the points raised during the review process.

Please, address and take into account all reviewers comments. Reviewer2 has numerous suggestions that will significantly improve the quality and impact of your manuscript, and Reviewer1 raises some fundamental points that should be addressed for acceptance of the paper.

Moreover, I, as the Editor, would like to ask you to justify that "light did not modify fish behavior" (Methods), i.e. state what type of controls have been performed to support this statement and to rule out multimodal control of behavior (visual system-lateral line interactions).

Please submit your revised manuscript within 2 months. If you will need more time than this to complete your revisions, please reply to this message or contact the journal office at plosone@plos.org. Please include the following items when submitting your revised manuscript:

We look forward to receiving your revised manuscript.

Kind regards,

Sylvie Rétaux

Academic Editor

PLOS ONE

Journal Requirements:

'The authors declare no competing or financial interests.'

We note that one or more of the authors are employed by a commercial company: Commercial Space Technologies, Ltd.

Additional Editor Comments (if provided):

Reviewers' comments:

Reviewer's Responses to Questions

**Comments to the Author**

1. Is the manuscript technically sound, and do the data support the conclusions?

Reviewer #1: Partly

Reviewer #2: Yes

2. Has the statistical analysis been performed appropriately and rigorously? 

Reviewer #1: No

Reviewer #2: N/A

3. Have the authors made all data underlying the findings in their manuscript fully available?

Reviewer #1: No

Reviewer #2: Yes

4. Is the manuscript presented in an intelligible fashion and written in standard English?

Reviewer #1: Yes

Reviewer #2: No

5. Review Comments to the Author

Reviewer #1: 1. The study presents the results of the original research.

This study describes a new insight into the horn of Chinese cavefish, which receive much attention but does not have a conclusive explanation of its function based on the experiments. This study first provided solid evidence of a hydrodynamic advantage of this hone.

2. Results reported have not been published elsewhere.

As far as I know, there is no comparative study based on Chinese cavefish.

3. Experiments, statistics, and other analyses are performed to a high technical standard and are described in sufficient detail.

The statistical part is unclear. How many individuals did this study used (Exp1 and Exp2)? How many were the casted models generated based on the different individuals of each of two species (how many biological replicates)? In other words, the sample size must be shown. As I assume the authors are from the engineering field, the sample size can be N=1 per species. If so, it should be noted so.

Other technical standards, such as PIV imaging, animal behavior tracking, and CFD modeling, seemed to provide detailed explanations.

4. Conclusions are presented properly and are supported by the data.

The authors conducted appropriate experiments and computer simulations to draw their conclusion; however, further data is needed to support it. Their major conclusion on the horn advantage is to create the low-pressure area around the cranial region.

(1) They conclude that canal neuromasts are located there, but I cannot clearly see them in Fig. 8B and 8C due to the low resolution of the images. It is also hard to believe the canal neuromast locations at the nose to head region: only a single line on the dorsal top of the cranium or bilateral?

(2) Authors are also ambiguous about the arguments between superficial neuromast and canal neuromast and how they gain sensitivity due to the horn structure. For example, at L501, “slow swimming behavior may improve the sensory capabilities of the lateral line system, in particular the superficial neuromasts.” But, at L525, “the buttressed head horn structure in its entirety enhances perception of pressure changes by the canal neuromasts (Fig 8E).”

The conclusion of L525 was, I assume, made because the authors only measured the pressure differences and the canal neuromasts but not superficial neuromasts are the pressure sensors (superficial neuromasts are for flow sensing). Authors need to aware that flow sensors can be stimulated between different pressures, although they are unlikely sensitive as the level as the canal neuromasts are (different pressures generate flow) – dipole pressure field generated by a vibrating rod could be detected by superficial neuromast (Yoshizawa et al., 2012., BMC Biol 2013, 11:82).

Accordingly, please describe and address the (potential) sensitivity gains of both the superficial and canal neuromasts

5. The article is presented in an intelligible fashion and is written in standard English.

Yes

6. The research meets all applicable standards for the ethics of experimentation and research integrity.

Yes

7. The article adheres to appropriate reporting guidelines and community standards for data availability.

No. The authors should upload their scripts used in these analyses, such as to Github and raw data for behavior-tracking, PIV analysis, to the public database (ex., Zenodo) to make them available to the public.

Other minor points:

L80: Authors should describe how they perform the 3-D rendering as other researchers can repeat their experiment.

L117: “there have been numerous observational, live-specimen studies of the sensory capabilities of the Mexican cave characin [29-43].” Please consider including a significant sensitivity work of (Yoshizawa, M., et al. (2014). The sensitivity of lateral line receptors and their role in the behavior of Mexican blind cavefish (Astyanax mexicanus). J. Exp. Biol. 217, 886–895.)

L301: With my expertise, I don’t understand “The CFD control volume was set to be ten times the fish’s body size (…) to ensure that the flow field was fully developed.” I hope other reviewer(s) look into this point.

L532: the discussion is a little too much, such as a statement about the deep-sea fishes, whose discussion does not conclude clearly. Please review and make this section (Horn function and other horn-bearing fish species) simpler and clearer.

Reviewer #2: This is an interesting paper on the potential role of the conspicuous horn-like structure on the head of a Chinese cavefish. The authors analyzed videos of the swimming behavior of the fish to obtain data on swimming style and velocity and combined this with 3D-scans of the fish and with Computational Fluid Dynamics simulations to support the notion that the horn alters the flow around the fish and thus the magnitude of the perceived stimulus at distinct locations on the fish surface. Although this is intuitively expected, it needs to be shown, and this is what the authors did. It appears that all experiments and analyses were properly performed and that the data are solid. The only problem with the manuscript is the style of writing which makes it in places hard to follow. I have attached a lin-by-line list of comments.

Title

Is too general and promises more than the paper can keep up with. Try to find a title that is more adequate and more precisely addresses the content of the work performed.

Abstract

Needs to be streamlined and brought to the point.

Line 20,23: Avoid abbreviations (CFD, PIV) in the Abstract. Rather spell out.

Line 21: Explain the term “other” hydrodynamic aspects.

Line 23: PIV is not particle image “velocity”. It is “velocimetry”.

Line 25: Remove “carefully-constructed”. The reader assumes that your science was carefully conducted.

Line 25: What type of simulations were done? What was simulated? Water flow? Then say so.

Line 26: What are “analogous simulations”.

Line 27: What is a “typical velocity”?

Line 27: Velocity and 3-D trajectories may be “features” of the swimming behavior, but “drag coefficients” and “sensing distance” are not. These are estimates derived from the simulations, right?

Line 38: “positively-correlated”. How can one correlate a single morphological aspect to anything? Rather say that this particular cranial morphology may lead to or provide greater sensitivity compared to fish that do not have such a morphology (if this is what you mean).

Introduction

Needs to be streamlined and brought to the point.

Lines 49-54 and Figure 1: I do not see the connection of the content of this paragraph to the main aspects of the paper, i.e., the horn and its role in sensing. I suggest to leave this out.

Line 64: “subdivided” instead of “bifurcated”

Line 72: Unclear. What is a “overall sculpted morphological form”? Do oyu want to say that all troglobite species have such a horn?

Line 86: “hyperthrophied” What exactly do you mean? In which way are the eyes hypertrophied? And in comparison to what? The eyes of troglophile species?

Line 90: I suggest to refer to Figure 2C here instead of line 93 where the figure does not really fit.

Line 95: “as opposed to” So there are no canal neuromasts?

Line 96: “hyperthrophied” See above: What exactly do you mean? In which respect and compared to what?

Lines 96-99: Do you have data on this, e.g., DASPEI stainings? If yes, it would be great to show.

Line 101: Reference 15 is not appropriate in this context. This is a modelling study and not a morphology paper.

Line 102: Context of this half-sentence to previous half-sentence not clear.

Line 107: remove “of course”.

Line 111-120: Context of this chapter on Astyanax to present study not clear. I suggest tp spare this for the Discussion.

Line 129: Strictly speaking, a morphological structure cannot directly enhance the sensitivity of a neuromast per se. As a matter of fact, this is not even necessary. A morphological structure can, however, affect the water flow in such a way that the stimulus to the sensor gets stronger, for instance by directing the flow to the sensor or by exposing a sensor that is located on the structure more to the flow.

Lines 131-139: Again, the context of this chapter to the present work is not evident and can be omitted.

Lines 140-172 (end of introduction): This part contains mainly Methods. By moving most of this to the Method section, this part can be streamlined and shortened substantially.

Line 143: “carefully-configures” See comment above. No need to stretch this point.

Methods

Lines 181,183: As stated above, figure 1A and 1B can be removed because it does not contribute to the data.

Line 185: Why only average length? Give range of lengths of animals.

Lines 191ff: Give the number of animals from which you obtained 3-D scans?

Line 199: Figure 3 does not show experiments. It rather shows the work flow. In any case, the figure is highly unusual and near-incomprehensible without much more explanatory text. In my opinion it does not help the paper but rather makes things more complicated. Remove it.

Line 213: Were the 3 trials run with 3 different fish? Specify. If only one fish, were the data from the 3 trials comparable?

Line 214: Not clear why the analysis window was reduced to the first 2 hours. Why measure 3 hours if you do not show the data?

Line 217 and fig. 4: What was the orientation/angle of view of the camera? From top, side, left, right? Please specify

Line 228: Line 213: Were the 6 trials run with 6 different fish? Specify. If only one fish, were the data from the 6 trials comparable?

Line 229: What was the particle diameter?

Line 232: Delete “the”

Line 232 ff: What type (model, manufacturer) of laser was used?

Line 242: What is an “ANN (U-net)”?

Results

Line 350-357: From how many fish were swimming data obtained? Did all fish show the same type of behavior? Were distances to wall and swimming velocities the same for these fish? Did all fish show the same burst and coast behavior?

Line 352 and Fig. 5A-D: Avoid extensive sub-divisions of figures. Here, A, B, C and D can be removed since each sub-figure has its own header.

Line 357 and Fig. 5F: Apparently, Fig. 5F shows something different from Figs. 5A-E. I suggest to make this a separate figure and relabel Figs. 5A-D as Fig. 5A and Fig. 5E as Fig. 5B.

Line 376: Chapter title unclear: “experiment 2 versus CFD”. This sound illogical.

Line 384: must be “were similar to …”

Line 385-386: Make sentences simpler by deleting “the morphology of” and “does that of”

Line 396: The red star in Fig. 6 appears in a boxplot. Thus, I assume it shows the median and not the mean.

Line 399: What is a “regressive best fit”? Do you mean that a linear regression was fitted to the data?

Line 404: “that” instead of “the!

Line 413: I am confused by this sentence. It is evident from Fig. 7C stage 3-M3 that the water is NOT quiet (see lower right side of figures with water velocities up to 2,8 cm/s). Please reconcile.

Line 417 and Fig. 7: The Figure is difficult to understand without further explanation.

Fig. 7A, X-axis: Is this distance of the fish to the laser plane? Why does it take 0.2 body lengths for the fish to enter the laser plane? Should this not be a single point at which the fish head touches the laser plane?

How long does it take for water velocity to decrease completely to pre-entering values?

Fig. 7B, Y-axis “Sense ability” is a strange term. Should this be sensing distance?

Lines 425-448: I assume that the canal pores drawn into Figure 8 are virtual pores. I suggest to state this again here.

Discussion

Line 450: Include a reference for the 1st sentence.

Line 452-457: What is the context of this part to the paper? One ´might as well leave this out and start the entire Discussion on line 457 with “Understanding the …”

Line 471: is comparable “to” that

Lines 503-510: Refer to and cite previously published data with similar findings for Astyanax.

Lines 511-531: What would be the driving selective force for developing such a horn for hydrodynamic reasons? What would be the stimulus that the fish is adapted to detect with this horn?

6. PLOS authors have the option to publish the peer review history of their article (what does this mean?). If published, this will include your full peer review and any attached files.

Reviewer #1: **Yes: **Masato Yoshizawa

Reviewer #2: No

---

## [Author Response · Author response to Decision Letter 0]

11 Nov 2021

Reply to Reviewer #1

Dear Dr. Yoshizawa,

Thank you very much for your time involved in reviewing the manuscript. Hereby, we try our best to revise the manuscript according to your comments and suggestions!

Comment 1: 

“This study describes a new insight into the horn of Chinese cavefish, which receive much attention but does not have a conclusive explanation of its function based on the experiments. This study first provided solid evidence of a hydrodynamic advantage of this horn.”

Response：

We appreciate your encouraging comment on the merit of this study. We hope to approach further in our following research. 

Comment 2:

The statistical part is unclear. How many individuals did this study used (Exp1 and Exp2)? How many were the casted models generated based on the different individuals of each of two species (how many biological replicates)? In other words, the sample size must be shown. As I assume the authors are from the engineering field, the sample size can be N=1 per species. If so, it should be noted so. 

Other technical standards, such as PIV imaging, animal behavior tracking, and CFD modeling, seemed to provide detailed explanations 

Response:

Thank you for your questions on the statistical part. We have added the relevant content in the revision, including the following details:

We have observed over ten individuals, and use one individual in Exp 1 and Exp 2 for quantify analysis. In Exp 1, we set 3 trials, and in Exp 2, we set 6 trials. And for CFD simulation, we chose the biggest samples to build the 3-D digital model.

We noted this information in [ For experiment 1 line 189-191: Experiment 1 was designed to observe the free swimming behavior of S. rhinocerous when introduced into a new environment and to estimate the average swim speed (Fig 2). Three 3-hr trials were performed on the same fish.

For experiment 2 line 203-209: Experiment 2 was designed to observe the swimming trajectories of the fish and to enable measurements of the flow field around the fish. The latter were used to empirically-estimate the drag coefficient and the potential maximum sensing distance (Fig 2). Six 3-hr replicate trials were performed on the same fish; during each trial we placed the fish in a 40 × 8 × 8 (cm) water-volume tank with in situ tiny polystyrene tracer particles (with diameter 10-100 μm) and allowed to acclimate for two hours. All measurements were then made during the third hour. 

For 3-D model line 168-171: The specimens that died during the transportation and maintenance at the facility were preserved in 75% alcohol solution, and the biggest one sampled was scanned to obtain the digital modeling (Figs 1B and 1D) by the Insight3 (Open Technologies) instrument at high scanning resolution (<0.05 mm per pixel).]

About the technique:

The fish tracking system details listed in Supplementary S1 and can see patent 202010478461.5.

CFD modeling set are listed in Supplementary S2.

PIV imaging requires that the particles density within the image is enough and the particle movement can be watched clearly. Details are shown in [line 212-216: As the fish swam, PIV measurements were taken during the steady or coasting phase of swimming. A laser pulse (Lifang Technique company, China) operated continuously with the fish crossing the laser plane multiple times. The program MicroVec V2.0 System (Tianjin University) was used to generate 2-D velocity fields.]

Comment 3:

“The authors conducted appropriate experiments and computer simulations to draw their conclusion; however, further data is needed to support it. Their major conclusion on the horn advantage is to create the low-pressure area around the cranial region. 

(1) They conclude that canal neuromasts are located there, but I cannot clearly see them in Fig. 8B and 8C due to the low resolution of the images. It is also hard to believe the canal neuromast locations at the nose to head region: only a single line on the dorsal top of the cranium or bilateral?

(2) Authors are also ambiguous about the arguments between superficial neuromast and canal neuromast and how they gain sensitivity due to the horn structure. For example, at L501, “slow swimming behavior may improve the sensory capabilities of the lateral line system, in particular the superficial neuromasts.” But, at L525, “the buttressed head horn structure in its entirety enhances perception of pressure changes by the canal neuromasts (Fig 8E).”

The conclusion of L525 was, I assume, made because the authors only measured the pressure differences and the canal neuromasts but not superficial neuromasts are the pressure sensors (superficial neuromasts are for flow sensing). Authors need to aware that flow sensors can be stimulated between different pressures, although they are unlikely sensitive as the level as the canal neuromasts are (different pressures generate flow) – dipole pressure field generated by a vibrating rod could be detected by superficial neuromast (Yoshizawa et al., 2012, BMC Biol 2013, 11:82). Accordingly, please describe and address the (potential) sensitivity gains of both the superficial and canal neuromasts”

Response:

Thanks for your comments and suggestions on the conclusion. To improve the clarity, we have made the following changes: 

(1) we adjust the neuromasts distribution figures’s scale as follows. The real neuromasts location of S. rhinocerous are shown in Fig. 8A after stained by immersion in DASPEI, that is, 2-(4-(dimethylamino)styryl)-N-ethylpyridinium iodide (DASPEI, Invitrogen), (Jørgensen, 1989). And in Fig8B we used a line to represent the supraorbital canal (so),

supratemporal canal (st) and trunk canal (tc). This simple method is similar with the previous study (Windsor, 2010a). But we also know this can’t represent the canal neuromasts precisely, we point this shortage in future plan. Please see details in [Line 597-601]

Jørgensen, J.M. (1989). Evolution of octavolateralis sensory cells. In The Mechanosensory Lateral Line (ed. S. Coombs, P. Görner, and H. Münz), pp. 115-145. New York: Springer-Verlag.

(2) Thank you for pointing out the difference between canal neuromast and superficial neuromast. In line 501 the ‘superficial neuromast’ should be the ‘canal neuromast’, and we have revised. Details in [lines 480-482: In addition, slow swimming behavior may improve the sensory capabilities of the lateral line system, in particular the canal neuromasts.]

(3) In this study, we mainly figure out the sensitivity gains of the canal neuromasts. For the superficial neuromasts, as you see, they sense the dipole pressure field generated by a vibrating rod could be detected by superficial neuromast. It need to set a different CFD simulation, we may need more precise simulation in future.

Comment 4:

“The authors should upload their scripts used in these analyses, such as to Github and raw data for behavior-tracking, PIV analysis, to the public database (ex., Zenodo) to make them available to the public”

Response:

Sorry, the analysis code can’t share in Githup now. 

The PIV analysis can be accomplished using free code named PIVlab in Matlab software or some economic software. In this study, we used the latter.

For behavior tracking, you can reference our patent 202010478461.5

The raw data for behavior tracking is video, It is too big to upload to Zenodo, but we will try to translate it into another format and share it in future. 

Comment 5:

“L80: Authors should describe how they perform the 3-D rendering as other researchers can repeat their experiment.

L117: “there have been numerous observational, live-specimen studies of the sensory capabilities of the Mexican cave characin [29-43].” Please consider including a significant sensitivity work of (Yoshizawa, M., et al. (2014). The sensitivity of lateral line receptors and their role in the behavior of Mexican blind cavefish (Astyanax mexicanus). J. Exp. Biol. 217, 886–895.)

L301: With my expertise, I don’t understand “The CFD control volume was set to be ten times the fish’s body size (…) to ensure that the flow field was fully developed.” I hope other reviewer(s) look into this point.

L532: the discussion is a little too much, such as a statement about the deep-sea fishes, whose discussion does not conclude clearly. Please review and make this section (Horn function and other horn-bearing fish species) simpler and clearer.”

Response:

L80: the 3-D digital model was rendered by 3-D scanning technology, we accomplish it with the help of a company, the details are shown in [lines 168-173: The specimens that died during the transportation and maintenance at the facility were preserved in 75% alcohol solution, and the biggest one sampled was scanned to obtain the digital modeling (Figs 1B and 1D) by the Insight3 (Open Technologies) instrument at high scanning resolution (<0.05 mm per pixel). These digital models were later used in CFD simulations to investigate the hydrodynamics of water flow around the fish.]

L117: Thank you for your suggestion, Yoshizawa et al., 2014 help us much with the undersanding of the lateral line system. And we have referenced it.

L301: in the simulation, the water flow will be influenced by the fish body and the wall. The CFD control volume was set to be ten times the fish’s body size to make sure that the calculation zone is big enough to simulation the case that the water flow around the fish body will not be influenced by the wall.

L532: We simple the discussion on the horn function. In details, we delete the discuss on sea-fish and some part about cavefish conservation in discussion to make the discussion section simpler.

 

Reply to Reviewer #2

Dear reviewer,

Comment 1:

“This is an interesting paper on the potential role of the conspicuous horn-like structure on the head of a Chinese cavefish. The authors analyzed videos of the swimming behavior of the fish to obtain data on swimming style and velocity and combined this with 3D-scans of the fish and with Computational Fluid Dynamics simulations to support the notion that the horn alters the flow around the fish and thus the magnitude of the perceived stimulus at distinct locations on the fish surface. Although this is intuitively expected, it needs to be shown, and this is what the authors did. It appears that all experiments and analyses were properly performed and that the data are solid. The only problem with the manuscript is the style of writing which makes it in places hard to follow. I have attached a lin-by-line list of comments.”

Response: 

Thank you very much for your time involved in reviewing the manuscript and your very encouraging comments on the merits. Your clear and detailed feedback help us much to improve this manuscript.

Comment on title2:

Title “Title is too general and promises more than the paper can keep up with. Try to find a title that is more adequate and more precisely addresses the content of the work performed”

Response:

‘Cavefish Hydrodynamics and Behavior: A Study of the Chinese cavefish Sinocyclocheilus rhinocerous’ has been changed into ‘Cavefish Hydrodynamics and Behaviors of the Chinese cavefish Sinocyclocheilus rhinocerous which possesses a head horn structure’

Comment on abstract:

Needs to be streamlined and brought to the point.

Line 20,23: Avoid abbreviations (CFD, PIV) in the Abstract. Rather spell out.

Line 21: Explain the term “other” hydrodynamic aspects.

Line 23: PIV is not particle image “velocity”. It is “velocimetry”.

Line 25: Remove “carefully-constructed”. The reader assumes that your science was carefully conducted.

Line 25: What type of simulations were done? What was simulated? Water flow? Then say so.

Line 26: What are “analogous simulations”.

Line 27: What is a “typical velocity”?

Line 27: Velocity and 3-D trajectories may be “features” of the swimming behavior, but “drag coefficients” and “sensing distance” are not. These are estimates derived from the simulations, right?

Line 38: “positively-correlated”. How can one correlate a single morphological aspect to anything? Rather say that this particular cranial morphology may lead to or provide greater sensitivity compared to fish that do not have such a morphology (if this is what you mean).

Response:

Line 20, 23: We spelled out CFD (Computational Fluid Dynamic) and PIV (particle image velocimetry). Please see details in [Lines 21-23: We use a combined approach of laboratory observations and Computational Fluid Dynamic (CFD) modeling to characterize the swimming behavior and other hydrodynamic aspects, i.e., drag coefficients and sensing distance of S. rhinocerous; Lines 23-26: Motion capture and tracking based on an artificial neural network, complemented by a particle image velocimetry (PIV) system to map out water velocity fields, were utilized to analyze the motion of a live specimen in a laboratory aquarium.] 

Line 21: The ‘other’ means ‘drag coefficients and sense distance’. Please see details in [Lines 21-23: We use a combined approach of laboratory observations and Computational Fluid Dynamic (CFD) modeling to characterize the swimming behavior and other hydrodynamic aspects, i.e., drag coefficients and sensing distance of S. rhinocerous].

Line 23: We have revised the ‘velocity’ into ‘velocimetry’, details in [Line 25, Line 126].

Line 25: We have removed the suggested content ‘carefully constructed’ and some similar descriptions like ‘carefully’. 

Line 25: We have explained the simulation type. Please see details in [Lines 26-28, CFD simulations on flow fields and pressure fields, based on digital models of S. rhinocerous, were also performed.].

Line 26: ‘analogous simulations’ means that we run the same simulation with S. rhinocerous on S. angustiporus.

Line 27: ‘Typical velocity’ means ‘average swimming velocities’. We have replaced the ‘typical velocity’ with ‘average swimming velocities’. Please see details in [Lines 30-33: Features of the cavefish swimming behavior deduced from the both live-specimen experiments and simulations included average swimming velocities and 3-D trajectories, estimates for drag coefficients and potential sensing distances, and mappings of the flow field around the fish.].

Line 27: The velocity and trajectories are the features of the swimming behaviors, but the “drag coefficients” and “sensing distance” are not. And the sensing distance are estimated from experiments, and the drag coefficients are estimates derived from both the simulations and experiments.

Line 38: We have revised the suggested content. Please see details in [Lines 39-41: Results support the hypothesis that the conspicuous cranial morphology of S. rhinocerous may lead to greater sensitivity compared to fish that don’t possess such morphology]

Comment on introduction:

Needs to be streamlined and brought to the point.

Lines 49-54 and Figure 1: I do not see the connection of the content of this paragraph to the main aspects of the paper, i.e., the horn and its role in sensing. I suggest to leave this out.

Line 64: “subdivided” instead of “bifurcated”

Line 72: Unclear. What is a “overall sculpted morphological form”? Do you want to say that all troglobite species have such a horn?

Line 86: “hyperthrophied” What exactly do you mean? In which way are the eyes hypertrophied? And in comparison to what? The eyes of troglophile species?

Line 90: I suggest to refer to Figure 2C here instead of line 93 where the figure does not really fit.

Line 95: “as opposed to” So there are no canal neuromasts?

Line 96: “hyperthrophied” See above: What exactly do you mean? In which respect and compared to what?

Lines 96-99: Do you have data on this, e.g., DASPEI stainings? If yes, it would be great to show.

Line 101: Reference 15 is not appropriate in this context. This is a modelling study and not a morphology paper.

Line 102: Context of this half-sentence to previous half-sentence not clear.

Line 107: remove “of course”.

Line 111-120: Context of this chapter on Astyanax to present study not clear. I suggest tp spare this for the Discussion.

Line 129: Strictly speaking, a morphological structure cannot directly enhance the sensitivity of a neuromast per se. As a matter of fact, this is not even necessary. A morphological structure can, however, affect the water flow in such a way that the stimulus to the sensor gets stronger, for instance by directing the flow to the sensor or by exposing a sensor that is located on the structure more to the flow.

Lines 131-139: Again, the context of this chapter to the present work is not evident and can be omitted.

Lines 140-172 (end of introduction): This part contains mainly Methods. By moving most of this to the Method section, this part can be streamlined and shortened substantially.

Line 143: “carefully-configures” See comment above. No need to stretch this point.

Respose

Line 49-54: We’ve deleted the Fig.1 and Lines 49-54. The paragraph one has been adjusted as [Over 150 species of cavefish have been discovered in China, accounting for over one third of the total number of cavefish species recorded worldwide [1-3]. Many cavefish species are at-risk or threatened, and functional extinctions likely have occurred or will occur even before numerous species are officially described [4, 5]. The overwhelming majority of Chinese cavefish species are classified as belonging to the genus Sinocyclocheilus (family Cyprinidae, sub-family Barbinae). Sinocyclocheilus cavefish display a stunning spectrum of distinctive adaptations to hypogean life unequaled by any other monophyletic cavefish group [6, 7].]

Line 64: We have changed the ‘bifurcated’ into ‘subdivided’. Please see details in [Line 54: Sinocyclocheilus cavefish can be subdivided into troglobite and troglophile species [8, 9].]

Line 72: ‘The overall sculpted morphological form’ means ‘the special body shape characteristics’, includes the head horn and humpback, not all troglobite species have such characteristics. We have revised the suggested content to the manuscript on [Lines 62-65: The humps and/or horns are components of the overall sculpted morphological form of the anterior and lateral body surfaces in some troglobite species such as S. rhinocerous (see Fig 1A).]

Line 86: ‘hypertrophied’ means ‘fat and big in size’, we have revised the suggested content to the manuscript on [Lines 74-76: They typically have pigment, eyes, bodies are hypertrophied in size and covered with particularly-fine scales (for a Barbinae species).].

Line 90: We refered to Fig. 1C (Fig 2C) here instead of line 93 as you suggested. Details in [Line 78-79: Superficially, troglophiles more closely resemble surface-dwelling (epigean) cyprinids (e.g., S. angustiporus – see Fig 1C)]

Line 95: The cavefish possesses both the canal neuromasts and superficial neuromasts. Here, we want to express that the S. rhinocerous possesses special superficial neuromasts distribution. We have deleted the “as opposed to”, please see details in [Lines 82-84: On its lateral body surfaces, S. rhinocerous possesses an extensive array of superficial neuromasts that extend from immediately behind the head until near the base of the tail.]

Line 96: We’ve deleted the “hyperthrophied”

Line 96-99: The DASPEI staining results are shown in Fig. 8B.

Line 101: we have replaced the reference 15 with the paper (Yoshizawa, M., et al. (2014). The sensitivity of lateral line receptors and their role in the behavior of Mexican blind cavefish (Astyanax mexicanus). J. Exp. Biol. 217, 886–895.).

Line 102: Context of this half-sentence to previous half-sentence not clear. The former half-sentence reviews the lateral line function; the later half-sentence reviews the mathematic analysis of the lateral line. Please Details in [lines 89-91: for lateral line structure and function review, see [13-15]; for earlier mathematical analyses of lateral line sensitivity, see [16-18]].

Line 107: we removed the “of course”.

Lines 131-139: we delete some contents about methods. Please see details in [Lines 120-134: In this paper, a combined approach of laboratory observations and CFD modeling was applied to characterize the behavior and swimming hydrodynamics of S. rhinocerous. The laboratory research consisted of observations of fish swimming in an aquarium. This utilized a motion capture and tracking system coupled to a particle image velocimetry (PIV) system to map out water-velocity fields. Motion capture and tracking are the primary techniques employed for many observations concerning fish swimming behavior [52]. In this study, a tracking system with an artificial neural network (ANN), Convolutional Networks for Biomedical Image Segmentation (U-Net) [45] was utilized to generate accurate trajectories (Patent 202010478461.5).]

Line 143: we deleted the “carefully-configures”

Comment on methods:

Lines 181,183: As stated above, figure 1A and 1B can be removed because it does not contribute to the data.

Line 185: Why only average length? Give range of lengths of animals.

Lines 191: Give the number of animals from which you obtained 3-D scans?

Line 199: Figure 3 does not show experiments. It rather shows the work flow. In any case, the figure is highly unusual and near-incomprehensible without much more explanatory text. In my opinion it does not help the paper but rather makes things more complicated. Remove it.

Line 213: Were the 3 trials run with 3 different fish? Specify. If only one fish, were the data from the 3 trials comparable?

Line 214: Not clear why the analysis window was reduced to the first 2 hours. Why measure 3 hours if you do not show the data?

Line 217 and fig. 4: What was the orientation/angle of view of the camera? From top, side, left, right? Please specify

Line 228: Line 213: Were the 6 trials run with 6 different fish? Specify. If only one fish, were the data from the 6 trials comparable?

Line 229: What was the particle diameter?

Line 232: Delete “the”

Line 232: What type (model, manufacturer) of laser was used?

Line 242: What is an “ANN (U-net)”?

Response

Line 181, 183: as suggested, we removed the Fig 1A and Fig 1B and adjust their description. Please see details in [lines 158-161: Most Chinese Sinocyclocheilus cavefish inhabit caves in the Yunnan-Guizhou Plateau, a well-developed karst landscape. Field surveys were carried out from 2015 to 2018, and S. rhinocerous and S. angustiporus specimens were obtained from several caves. The adult specimens of S. rhinocerous had a maximum body length of 10 cm (general body length 4-10 cm) (Fig 1A).]

Line 185: The 10 cm is the maximum body length, and we supplied the body length range (4-10 cm). Please see details in [line 161: The adult specimens of S. rhinocerous had a maximum body length of 10 cm (general body length 4-10 cm) (Fig 1A).]

Line 191: One S. rhinocerous and S. grahami samples were used to obtain the 3-D scans. Please see details in [lines 168-171: The specimens that died during the transportation and maintenance at the facility were preserved in 75% alcohol solution, and the biggest one sampled was scanned to obtain the digital modeling (Figs 1B and 1D) by the Insight3 (Open Technologies) instrument at high scanning resolution (<0.05 mm per pixel).]

Line 199: The Fig. 3 (Fig. 2 now) shows all the work in this paper and can help to understand the wok flow.

Line 213: All the three trails run with the same fish, and the results show all three trails show the same characteristics. Please see details in [Lines 189-192: Experiment 1 was designed to observe the free swimming behavior of S. rhinocerous when introduced into a new environment and to estimate the average swim speed (Fig 2). Three 3-hr trials were performed on the same fish.]

Line 214: we measure the behavior 3 hours to make sure the time is enough for fish to be familiar with the environment. And the results show that 2 hours is enough for the fish to be familiar with this experimental environment, so we only analysis the first two hours. Please see details in [Lines 191-193: It was found that the fish behaved differently in the first two hours and became stable in the third hour, thus we analyzed the swimming behavior during the first two hours.]

Line 217: Every time we rotate the image in 90°. Please see details in supplementary S1[lines 5-8: In Experiment 1, after lens correction, 220 top view photo images (Fig A1-A) and 220 side view images (Fig A1-B) were randomly selected. We then labeled the fish in each image (Figs. A1-A, B) and rotated the labeled results (Figs. A1-a, b) (90°, 180o and 270o).].

Line 228: All the three trails run with the same fish. And the results show that the six trails show the same characteristics. We supply the detail [Lines 203-210: Experiment 2 was designed to observe the swimming trajectories of the fish and to enable measurements of the flow field around the fish. The latter were used to empirically-estimate the drag coefficient and the potential maximum sensing distance (Fig 2). Six 3-hr replicate trials were performed on the same fish; during each trial we placed the fish in a 40 × 8 × 8 (cm) water-volume tank with in situ tiny polystyrene tracer particles (with diameter 10-100 μm) and allowed to acclimate for two hours. All measurements were then made during the third hour.]

Line 229: The particle diameter is 10-100μm, please see details in [Lines 206-209: Six 3-hr replicate trials were performed on the same fish; during each trial we placed the fish in a 40 × 8 × 8 (cm) water-volume tank with in situ tiny polystyrene tracer particles (with diameter 10-100 μm) and allowed to acclimate for two hours.]

Line 232: We have deleted “the”. 

Line 232: The lased is produced by Lifang Technique company, China. And the whole software was supported by Tianjin University, China. Please see details in [Lines 212-216: As the fish swam, PIV measurements were taken during the steady or coasting phase of swimming. A laser pulse (Lifang Technique company, China) operated continuously with the fish crossing the laser plane multiple times. The program MicroVec V2.0 System (Tianjin University) was used to generate 2-D velocity fields.]

Line 242: ANN is artificial neural network, U-net is one of the ANN and named “Convolutional Networks for Biomedical Image Segmentation”. Please see details in [Lines 126-129: In this study, a tracking system with an artificial neural network (ANN), Convolutional Networks for Biomedical Image Segmentation (U-Net) [45] was utilized to generate accurate trajectories (Patent 202010478461.5).]

Comment on results:

Line 350-357: From how many fish were swimming data obtained? Did all fish show the same type of behavior? Were distances to wall and swimming velocities the same for these fish? Did all fish show the same burst and coast behavior?

Line 352 and Fig. 5A-D: Avoid extensive sub-divisions of figures. Here, A, B, C and D can be removed since each sub-figure has its own header.

Line 357 and Fig. 5F: Apparently, Fig. 5F shows something different from Figs. 5A-E. I suggest to make this a separate figure and relabel Figs. 5A-D as Fig. 5A and Fig. 5E as Fig. 5B.

Line 376: Chapter title unclear: “experiment 2 versus CFD”. This sound illogical.

Line 384: must be “were similar to …”

Line 385-386: Make sentences simpler by deleting “the morphology of” and “does that of”

Line 396: The red star in Fig. 6 appears in a boxplot. Thus, I assume it shows the median and not the mean.

Line 399: What is a “regressive best fit”? Do you mean that a linear regression was fitted to the data?

Line 404: “that” instead of “the!

Line 413: I am confused by this sentence. It is evident from Fig. 7C stage 3-M3 that the water is NOT quiet (see lower right side of figures with water velocities up to 2,8 cm/s). Please reconcile.

Line 417 and Fig. 7: The Figure is difficult to understand without further explanation.

Fig. 7A, X-axis: Is this distance of the fish to the laser plane? Why does it take 0.2 body lengths for the fish to enter the laser plane? Should this not be a single point at which the fish head touches the laser plane? How long does it take for water velocity to decrease completely to pre-entering values?

Fig. 7B, Y-axis “Sense ability” is a strange term. Should this be sensing distance?

Lines 425-448: I assume that the canal pores drawn into Figure 8 are virtual pores. I suggest to state this again here.

Response

Line350-357: We have observed the fish swimming behavior in KIZ for around ten fishes. They show the same burst and coast behavior. And we use one fish in experiment for three trials. 

Line 352: we renamed the Fig. 5, removed the A, B, C, D and renamed them as A.

Line 357: we have separated the Fig. 5F 

Line 376: we have change the title as “experiment 2 and CFD simulation”

Line 384: we have revised it as “were similar to” [line 392]

Line 385-386: we have deleted the “the morphology of” and “does that of” as suggested.

Line 396: we have revised the “mean” as “median” [line 405]

Line 399: yes, we mean a linear regression was fitted to the data [lines 369-371: The simulation results from this study (not including the point based on Experiment 2) are fitted by the linear relation]

Line 404: we replaced the “the” with “that” as suggested [line 414]

Line 413: the water is not quiet, but it will be when the time is enough.

Line 417: Typically, E_Lwas initially low during the approaching stage (Fig 7A, stage 1), and the water velocity field was relatively smooth (Fig 7C, stage 1-M1). As the fish approached the laser plane, the E_Lvalue increased sharply, reaching a maximum value when the fish head touched (entered) the laser plane (Fig 7A, stage 2). This was accompanied by a disruption in the water velocities (Fig 7C, stage 2-M2).

Lines 417: we have revised the sense ability into sense distance.

Lines 425-448: we redraw the canal neuromasts distribution in Fig 8A.

Comment on discussion:

Line 450: Include a reference for the 1st sentence.

Line 452-457: What is the context of this part to the paper? One ´might as well leave this out and start the entire Discussion on line 457 with “Understanding the …”

Line 471: is comparable “to” that

Lines 503-510: Refer to and cite previously published data with similar findings for Astyanax.

Lines 511-531: What would be the driving selective force for developing such a horn for hydrodynamic reasons? What would be the stimulus that the fish is adapted to detect with this horn?

Response

Line 450: S. rhinocerous, like many cavefish species, is extremely vulnerable to perturbations in its habitat [61-64].

 Bitchuette ME, Trajano E. Conservation of subterranean fishes. In: Trajano M, Bitchuette E, Kappor BG, editors. Biology of Subterranean Fishes. New Hampshire, USA: Science; 2010. p. 65-80.

 Shu SS, Jiang WS., Whitten T, Yang JX, Chen XY. Drought and China’s cave species. Science. 2013;340:272.

 Poulson TL. Cavefish: retrospective and prospective. In: Trajano M, Bitchuette E, Kappor BG, editors. Biology of Subterranean Fishes. ew Hampshire, USA: Science; 2010. p. 1-40.

 Xing YC, Zhang CG, Fan EY, Zhao YH. Freshwater fishes of China: species richness, endemism, threatened species and conservation. Diversity and Distributions. 2016;22:358-370.

Line 452-457: the context of this part to the paper is for introduce that cavefish conservation is hard without knowledge of fish swimming behavior.

Line 471: we have added the ‘to’. Please see details in [lines 449-452: The sensing distance of 0.25 BL observed in the experiment (Fig 4B) is comparable to that reported for blind cave characins [40, 41], the blind hypogean morph of Astyanax mexicanus.]

Lines 503-510: There is no swimming velocity report on Astyanax mexicanus, but we compare the swimming velocity of S. rhinocerous with American cavefish [lines 467-469: and the average swimming speed was estimated as 1.42 cm/s for A. spelea, 0.72 cm/s for T. subterraneus, 0.45 for A. rosae, and 0.16 cm/s for F. agassizii [67]]. And we compare the wall-following behavior and sensing distance to Astyanax mexicanus [lines 449-452: The sensing distance of 0.25 BL observed in the experiment (Fig 4B) is comparable to that reported for blind cave characins [40, 41], the blind hypogean morph of Astyanax mexicanus.]

Lines 511-531: This is a very professional question. Unfortunately, there is no answer so far. Our study only discusses the influence of the head horn structure to the fishes’ drag coefficient and the lateral line stimulus. To answer this question, may need more studies from biologists. But we will also try our best to explore the truth.

---

## [Decision Letter · Decision Letter 1]

26 Jan 2022

PONE-D-21-12656R1Cavefish Hydrodynamics and Behaviors of the Chinese cavefish Sinocyclocheilus rhinocerous which possesses a head horn structurePLOS ONE

Dear Dr. Xu,

Thank you for submitting your manuscript to PLOS ONE. After careful consideration, we feel that it has merit but does not fully meet PLOS ONE’s publication criteria as it currently stands. Therefore, we invite you to submit a revised version of the manuscript that addresses the points raised during the review process.

ACADEMIC EDITOR: As you will see, the two reviewers have found very substantial improvements in your manuscript. Rev1 insists on the submission of raw data and code. Rev2 still has a list of suggestions on the writing and also on some scientific aspects of your study. Please, respond and change the manuscript according to all these comments. We will be happy to receive your second revised version shortly.

We look forward to receiving your revised manuscript.

Kind regards,

Sylvie Rétaux

Academic Editor

PLOS ONE

Reviewers' comments:

Reviewer's Responses to Questions

**Comments to the Author**

1. If the authors have adequately addressed your comments raised in a previous round of review and you feel that this manuscript is now acceptable for publication, you may indicate that here to bypass the “Comments to the Author” section, enter your conflict of interest statement in the “Confidential to Editor” section, and submit your "Accept" recommendation.

Reviewer #1: All comments have been addressed

Reviewer #2: (No Response)

2. Is the manuscript technically sound, and do the data support the conclusions?

Reviewer #1: Yes

Reviewer #2: Yes

3. Has the statistical analysis been performed appropriately and rigorously? 

Reviewer #1: Yes

Reviewer #2: I Don't Know

4. Have the authors made all data underlying the findings in their manuscript fully available?

Reviewer #1: No

Reviewer #2: Yes

5. Is the manuscript presented in an intelligible fashion and written in standard English?

Reviewer #1: Yes

Reviewer #2: Yes

6. Review Comments to the Author

Reviewer #1: The authors made significant efforts to improve this manuscript following both reviewers' comments. It is very nice to see this version. The authors still have a struggle to upload/share their raw data and analysis scripts/codes. I will ask the editor for looking into whether the PLoS One guideline is OK with it.

Reviewer #2: This is an improved version that is much more comprehensible than the previous one. However, there are still a number of minor, but also some more serious issues that need to be addressed and/or corrected.

Title:

Just another suggestion. Simply “Swimming Behavior and Hydrodynamics of the Chinese cavefish, Sinocychlocheilus rhinoceros” or maybe “Swimming Behavior and Hydrodynamics of the Chinese cavefish, Sinocychlocheilus rhinoceros, and a possible role of its distinct head horn structure”

Abstract

Abbreviations (CFD, PIV) are still present in the Abstract even though the definitions are spelled out. In most other journals this is not common. Abbreviations are normally given at the first site in the main text body where they are defined first, but not in an Abstract.

Line 25/26: “Particle Image Velocimetry”. Start each word with a capital letter.

Line 34: “sensing distance”. Do you mean sensing distance of the lateral line? If so, include this.

Line 36: Use past tense when reporting own data.

Line 36: If you include lateral line in line 34, then you should delete the text in parentheses here.

Line 44: The following comment applies to the statement that the sensitivity of sensing is increase in S.r. (i.e., here, in Methods, Results and Discussion). If I understand the paper correct, then there is greater pressure (pressure gradient) on the head of S. rhinoceros as compared to S. angustiporus. Thus, there is a stronger stimulus to this region in S.r. than in S.a. But this does not mean or show in any way that the neuromasts in this region have greater sensitivity. This can only be shown by comparing data from e.g., neuronal recordings from these neuromasts, or from observing neuromast motion in response to identical stimuli applied to both species. One could also do a histological survey of the neuromasts in that region and compare neuromast numbers, size, innervation etc. and from this conclude on sensitivity. The fact of a greater stimulus is no proof for greater sensitivity.

Introduction

The beginning of the Introduction is much clearer now due its shortness. However, the end of the Introduction can still be streamlined further. Presently there are numerous methodical aspects presented at the end of the Introduction, that should be reported in the Methods part (see comments below).

Line 85 and elsewhere in manuscript: There is no need any more to refer to the field survey (except at one site in the Discussion)

Line 102 ff: I appreciate the description of the lateral line morphology of S.r.. But (and perhaps I made this comment before), you nonetheless need to give the original reference here or instead show your own data. I checked references 13-15 but S.r. is not mentioned in any of these references.

Line 130: I suggest a line break here after (NACA0013).

Line 139: “increase the sensitivity”. Same comment as in Abstract line 44. There is a greater stimulus on the head but this is no proof of greater sensitivity of the receptors (see also comment above and elsewhere in this list).

Line 150 and also 136: You tend to repeat “This paper” or “In this paper”. Try to avoid these repetitions. E.g., here you can simply delete it. Also, lines 150-152 are more or less the same as lines 113-120. Go through the manuscript and remove such redundancies wherever they occur.

Lines 152 ff: I suggest to keep the end of the Introduction as straight-forward as possible by moving all the methodical details to the Method section. For example, one could simply say: “In this paper, a combined approach of laboratory observations including fish tracking, PIV and CFD modeling was used to characterize the swimming behavior and hydrodynamics of S. rhinocerous. The results of our data analysis suggests that …” All the rest, the pros and cons and the good and bads of your methods can go into the Methods section.

Materials and methods

Line 187: “field survey”. Not needed any more (see above).

Line 202: Must be specimen, not speciemen.

Line 206: “scanned to obtain digital modeling”. Do you mean “scanned for imaging and subsequent digital modeling”?

Line 211-215: It is too bad that behavioral data were collected on only a single fish. Results probably differ quite a bit when fish of different size were used. It would good to have a feeling for this variability.

Line 215: Should be specimen, not specimens since all experiments were done on a single fish.

Line 215: “not affect behavior”. What exactly is meant here? Which aspect of behavior? How do you know? How did you control for possible effects?

Line 225ff: Here you could include all the details of the tracking system.

Lien 232: I guess this is length x width x height? What was the water level? Also 8 cm? Or lower? This would then essentially be a 2D-pkane in which the fish was moving.

Line 236ff: Did camera 3 really have a frame rate of only 10 Hz? This then limits you PIV analysis to 10 Hz.

Lines 241ff: In this section you can include all the details on the PIV system.

Line 251: “A laser pulse operated continuously”. This sentence is ambiguous. Be more specific. Also, the information provided here is rather limited. Did you use a PW or a CW laser? I assume it was a PW laser that was turned on for the entire recording session? Please make this clear. Also, what laser model did you use? What was the wavelength (IR, red, green)? How did you produce the laser sheet? I assume you used a dispersion lens? What was your level of analysis (pixel size, interrogation window etc.)?

Line 261: “50 times per second”. If camera 3 was operating at 10 Hz, then you cannot be faster than this.

Line 308ff: I am not sure if I understand this correctly. You measure an abrupt change in energy across the laser plane caused by the approaching fish (Fig. 7A). This depends on swimming velocity. The faster the fish, the earlier you detect this. In other words, the faster the fish, the further the fish is away from the laser plane when you detect the disturbance (Fig. 7B, � increase in BL). I assume this is the case because the stimulus produced by the fish, most likely the bow wake, increases with swimming speed? But I still do not understand why this translates into sensing distance for the fish lateral line system? Please help the reader here. Do you want to say that the fish detects its own bow wake? But then why does it need 0,2 BL for this? Are there no neuromasts near the tip of the snout?

Line 317: It would be helpful if you make clear that this was a simulated flow that was continuously on and not a real flow in a flow channel. The reader may have forgotten or overlooked this here.

Line 352: “2% of body length”. Why do you make such assumptions? From your writing further below, I infer that you do have the real data on canal pore spacing. So why not use the real data?

Line 358: reference 50 is a book chapter by Jorgensen. While Jorgensen does refer to DASPEI stain as a method, I have not found any reference to S. rhinoceros in that chapter. As requested further up, I ask for the original reference or show you own data.

Results

Line 369: “more randomly”. There is no comparative to random. Also, how did you test for randomness?

Lines 368ff: You analyzed the data from the 3rd hour but you show the data from hours 1 and 2. Where are the data from the 3rd hour? Show!

Line 403: must be “shows”

Line 403ff: Do I understand correctly that Fig. 6A show data from one example? So what is the drag coefficient that results from this example?

Line 407: Can’t be the same Re value because data points in figure are not aligned on the X-axis.

Line 408: “consistent with that obtained from Experiment 2”

Line 409: “tended”. Why tended? I think it is very clear from the figure that they did decrease.

Line 410/411: “were similar to, but consistently higher than those …”

Line 422: “indicates”

Line 428 and subsequent paragraph: sensing distance. See my questions above.

Line 429ff: The description of the data still does not match the data shown. The text reads as if the velocity in stage 3 is similar to that in stage 1, which is evidently not the case. During stage 3, the velocity is still greater than during stage 1, and this should be described as such. The figure does not show if velocity decreased eventually to values similar to this in stage 2, but you can say this but have to make it clear.

Line 443ff: What part of the 40x8x8 cm tank is shown in the lower part of the figure? Is this a vertical or a horizontal laser plane? Can you draw the outlines of the fish into the lower figures?

Line 450: “Lateral stimulus”. Do you mean “Lateral line stimulus”?

Line 451 and figures 8 and 9: Apparently, part A of Figure 8 shows a DASPEI stain. You need to say this somewhere in the text or legend. You also need to indicate that the upper two photographs show a dorsal view and the lower two show a side view of the fish. By the way, are these from the same fish or from two different fish? Also, separate the upper two photographs as you did the lower two.

I assume the scale bar of 1 cm in the upper right photograph refers to all 4 photographs. This means that the upper magnification (dotted rectangle) is ca. 5x5 mm and the lower one ca. 3x2 mm.

Now, by looking at the magnifications (left photographs in 7A), I do see lines of dots extending from left to right on the head and from dorsal to ventral on the side. However, I do not see something like a trunk lateral line canal. Moreover, each line in the lower figure has at least 10, if not more dots, i.e., at least 5 dots per 2 mm. This all makes me assume that these are not canal neuromasts that are stained here, but rather superficial neuromasts. In any case, the spacing is much greater than the 2 mm pore distance assumed for the CFD analysis that is shown in part B of Figure 8. If the stained neuromasts in Fig. 8A are indeed superficial neuromasts, then showing this is not helpful. Instead, the reader needs to see a stain of the canals, the canal pores and/or the canal neuromasts. To show the canals and the pores, one could simply fill the canal with methylene blue to visualize its course. The pores and the canal neuromasts normally also are stained in a DASPEI stain, but in Fig. 8A I do not see them. But it is necessary for your conclusions to show that indeed there are canal neuromasts in that region.

Line 482, figure legend Fig. 9: “Lateral line stimulus”? You may also point out here, that the Y-axis is plotted reversed (from negative to positive values).

Figures 8-9: At some point you should explicitly write, how you come from the data shown in Fig. 8B to the curves shown in Fig. 9A. I assume that you plotted normalized pressure along the putative canal neuromasts (dots in Fig. 8B)? But you do not explain this. Help the reader here.

Figure 9: I compared that curves in Fig. 9A with those in Fig. 7 by Windsor et al (2010a). While the blue curve seems to be the curve from the Hassan-data. The green curve seems to be a curve that is derived from a NACA airfoil. The actual 2D fish data look different. Check red curve in Fig. 7 by Windsor et al). It actually does have two negative peaks but the positive peak in between is very much smaller than the one here in S. rhinoceros.

Also, I recommend to label all curves with the species name (A. mexicanus in both cases) and give the respective references in parentheses).

Discussion

Line 495-496: The sentence reads odd. Maybe you need to shorten this to “Understanding the …. may help to discern …”

Line 509: Unclear. Do you mean that the wall-following phase is the exploratory phase?

Line 525: Delete text in parenthesis. You explain the methods a few lines further below.

Line 545-546: slow swimming may improve sensory capabilities. Why? Give a rational or a logic why this is so.

Line 553: “It is conjectured“. Do you want to say “We therefore hypothesize…”?

Line 563: pressure fields matched the locations of the canal pores. Unclear description. Be more explicit. What exactly do you mean?

Line 573: “It is conjectured“. Do you want to say “We propose that…”? After all, the data do not and cannot show this.

Line 607: Why should this not have a similar function. If the form is similar to the form in S.r., then one would expect similar hydrodynamic effects. And as you hypothesize yourself, better sensing may lead to increased fitness, even if thus a trait only found in males.

Lines 656ff: I really think that the future plans can be omitted. By all due respect for your research, but it is really not important for the present paper what you want to do next. If these plans contain arguments relevant for the interpretation of the present data, then these arguments can be discussed in those sections of the Discussion where they are appropriate. If this future research is essential for the data, then maybe one should perform this research now and include it in the paper.

7. PLOS authors have the option to publish the peer review history of their article (what does this mean?). If published, this will include your full peer review and any attached files.

Reviewer #1: **Yes: **Masato Yoshizawa

Reviewer #2: No

---

## [Author Response · Author response to Decision Letter 1]

21 Mar 2022

Dr. Yoshizawa, We have upload all analysis data. But, We are sorry, the analysis code still can’t share in Githup now. For that we have accomplished the PIV analysis using an economic software. For behavior tracking, we have developed a tracking system, but we have applied a patent, if you have interest in it in future, you can search our patent 202010478461.5.

For review 2

Dear reviewer， thank you very much for your time involved in reviewing the manuscript and your very encouraging comments on the merits. Your clear and detailed feedback help us to improve the present study much..

---

## [Editor Report · Decision Letter 2]

11 Apr 2022

PONE-D-21-12656R2Swimming Behavior and Hydrodynamics of the Chinese cavefish Sinocyclocheilus rhinocerous and a possible role of its head horn structurePLOS ONE

Dear Dr. Xu,

Thank you for submitting the second revision of your manuscript to PLOS ONE.

Please know that PLOS ONE requires that all code be made available without restrictions upon publication of the work in the Supporting Information files or in a publicly available repository. You may view the following link for more information: https://journals.plos.org/plosone/s/materials-software-and-code-sharing. 

Additionally, it looks like the authors are giving an unacceptable restriction regarding the availability of their data. We do not allow authors to restrict their data because of personal restrictions, such as patents or future publications. For more information, please see: https://journals.plos.org/plosone/s/data-availability#loc-unacceptable-data-access-restrictions.

Therefore, if you do not wish to comply with these requirements, I am sorry to say that PlosOne will have to reject your manuscript.

We look forward to receiving your revised manuscript.

Kind regards,

Sylvie Rétaux

Academic Editor

PLOS ONE
---

## [Author Response · Author response to Decision Letter 2]

14 Jun 2022

Thank you for your suggestions, we have upload the data and the code in supporting information.

---

## [Editor Report · Decision Letter 3]

22 Jun 2022

Swimming Behavior and Hydrodynamics of the Chinese cavefish Sinocyclocheilus rhinocerous and a possible role of its head horn structure

PONE-D-21-12656R3

Dear Dr. Xu,

We’re pleased to inform you that your manuscript has been judged scientifically suitable for publication and will be formally accepted for publication once it meets all outstanding technical requirements. I also wish to sincerely thank the two reviewers for their help in improving the manuscript.

Kind regards,

Sylvie Rétaux

Academic Editor

PLOS ONE

Additional Editor Comments (optional):

The full code for analysis has been submitted as Supplemental material, therefore the manuscript is now conform to PlosOne policies; and the changes suggested by reviewer 2 have been added.
---

## [Editor Report · Acceptance letter]

29 Jun 2022

PONE-D-21-12656R3 

Swimming Behavior and Hydrodynamics of the Chinese cavefish *Sinocyclocheilus rhinocerous* and a possible role of its head horn structure 

Dear Dr. Xu:

I'm pleased to inform you that your manuscript has been deemed suitable for publication in PLOS ONE. Congratulations! Your manuscript is now with our production department. 

Kind regards, 

on behalf of

Dr. Sylvie Rétaux 

Academic Editor

PLOS ONE